# Mortality risks from a spectrum of causes associated with sand and dust storms in China

Can Zhang[1,3], Meilin Yan[2,3], Hang Du[1], Jie Ban[1], Chen Chen[1], Yuanyuan Liu[1] & Tiantian Li [1]✉

Sand and Dust Storms (SDS) pose considerable health risks worldwide. Previous studies only indicated risk of SDS on overall mortality. This nationwide multicenter time-series study aimed to examine SDS-associated mortality risks extensively. We analyzed 1,495,724 deaths and 2024 SDS events from 1 February to 31 May (2013–2018) in 214 Chinese counties. The excess mortality risks associated with SDS were 7.49% (95% CI: 3.12–12.05%), 5.40% (1.25–9.73%), 4.05% (0.41–7.83%), 3.45% (0.34–6.66%), 3.37% (0.28–6.55%), 3.33% (0.07–6.70%), 8.90% (4.96–12.98%), 12.51% (6.31–19.08%), and 11.55% (5.55–17.89%) for ischemic stroke, intracerebral hemorrhagic stroke, hypertensive heart disease, myocardial infarction, acute myocardial infarction, acute ischemic heart disease, respiratory disease, chronic lower respiratory disease, and chronic obstructive pulmonary disease (COPD), respectively. SDS had significantly added effects on ischemic stroke, chronic lower respiratory disease, and COPD mortality. Our results suggest the need to implement public health policy against SDS.

Sand and Dust Storms (SDS) are emerging as a considerable public health concern. SDS were estimated to contribute to up to 2 billion tonnes of dust emissions into the air annually, resulting in poor air quality for over 150 countries[1]. In recent decades, the changes in climate and land use cover have intensified the frequency and intensity of SDS[2,3]. In 2021, the World Health Organization (WHO) announced the updated global air quality guidelines;[4] WHO has long wanted to address SDS in this update, but the insufficient and unspecific epidemiological evidence on SDS hampers it[4].

Existing studies have indicated the overall effects of SDS on circulatory and respiratory mortality[5–7]. A recent meta-analysis estimated a 2.33% pooled increase in circulatory mortality on days with SDS and a 3.99% in respiratory mortality on three days after SDS[6]. However, little is known about SDS impact on mortality due to other diseases, including circulatory and respiratory sub-causes. Also, most relevant studies were conducted at a single location with relatively small sample

sizes[8–11]. The lack of a national assessment of SDS health impact impedes the science-based national and regional cooperation to mitigate and cope with the adverse effects of SDS[12].

Inhalable particulate matter ($PM_{10}$), constituted of fine particulate matter ($PM_{2.5}$) and coarse fine particulate matter ($PM_{2.5-10}$), is well known to be the main component of SDS, posing a threat to human health[13]. For example, Neophytou and colleagues observed that a 10 μg/m³ increase in $PM_{10}$ concentration during SDS was associated with a 2.43% increase in cardiovascular mortality[14]. Moreover, there is evidence that heavy $PM_{2.5}$ pollution events increased mortality risks and caused an independently added effect[15]. Whether SDS events, a special kind of particulate matter (PM) pollution event with high concentration of $PM_{2.5-10}$ and $PM_{2.5}$, have added effects on mortality is yet to be determined.

Here, we conducted a nationwide multicenter time series study in China. Our objectives were to: (1) investigate the overall short-term

[1]China CDC Key Laboratory of Environment and Population Health, National Institute of Environmental Health, Chinese Center for Disease Control and Prevention, Beijing, China. [2]School of Ecology and Environment, Beijing Technology and Business University, Beijing, China. [3]These authors contributed equally: Can Zhang, Meilin Yan. ✉e-mail: litiantian@nieh.chinacdc.cn

effects of SDS events on mortality from a series of causes, identifying the spectrum of SDS-sensitive health outcome; (2) explore the added short-term effects of SDS events on mortality. Findings from this study will improve current understanding of the health effects of SDS.

## Results

### Summary statistics for SDS events and mortality

From 2013 to 2018, 2,024 SDS events were identified in 214 Chinese counties during the SDS period (1 February–31 May), with a high frequency of SDS events (Table 1, Fig. S1). In general, SDS events occurred more frequently in counties located in the northern parts of China that are close to the dust sources than in counties in southern regions (Fig. 1). A total of 1,495,724 deaths occurred during the SDS period, and 19,082 deaths in the identified SDS event days (Table 2). Ischemic heart disease mortality accounted for most circulatory mortality (40.26%), and chronic lower respiratory disease accounted for most respiratory mortality (71.84%).

### Association between SDS events and mortality

The estimated pooled effects of SDS events on a spectrum of mortality outcomes at lag day 0 are shown in Fig. 2. We observed increased risks associated with SDS events for most of the mortality outcomes. For mortality of broad category causes, respiratory disease showed the highest increased risk (8.90%; 95% confidence interval [CI]: 4.96%, 12.98%) during SDS event days compared with clean days. SDS events were also associated with increased circulatory mortality, genitourinary mortality, nervous disease mortality, and digestive disease mortality, though the effect estimates were not statistically significant. For cause-specific mortality, SDS event exposures were significantly associated with elevated mortality risk for ischemic stroke, intracerebral hemorrhagic stroke, hypertensive heart disease, myocardial infarction, acute myocardial infarction, acute ischemic heart disease, chronic lower respiratory disease, and chronic obstructive pulmonary disease (COPD); ischemic stroke and chronic lower respiratory disease presented the highest increased risk, with the excess risk (ER) of 7.49% (95% CI: 3.12%, 12.05%) and 12.51% (6.31%, 19.08%) among sub-causes of circulatory and respiratory diseases, respectively. The results of the delayed effect of SDS are provided in the Supplementary Information (Fig. S2). And the highest and most significant mortality risks from most diseases were observed at day lag0. Similar effect estimates were also observed at lag 1 day, while more negligible effects or no associations at the next 2–3 days (Fig. S2).

Compared to clean days, we observed a more enhanced risk of mortality for ischemic stroke, intracerebral hemorrhagic stroke, hypertensive heart disease, chronic lower respiratory disease, and COPD on SDS event days than non-SDS event days with $PM_{2.5}$ pollution (Fig. S3). With adjustments of $PM_{2.5-10}$ and $PM_{2.5}$ in the regression models, the added effects of SDS events on mortality were generally less substantial than the overall effects of SDS events (Fig. 3). Despite this, we still observed significantly increased risks associated with SDS events for ischemic stroke, chronic lower respiratory disease, and COPD, suggesting added effects of SDS events on these mortality outcomes (Fig. 3).

In analyzing SDS-associated mortality risk stratified by sex and age group, we found a more pronounced risk of respiratory mortality for males than females and the older than the younger (Fig. S4). By contrast, females presented slightly higher risk than males and the younger higher than the older, for all-cause, non-accidental, and circulatory mortality. We did not find significant differences in the estimates of SDS-associated mortality risk by sex and age group (Fig. S4).

Under more lenient SDS events definitions, with a higher $PM_{2.5}$/$PM_{10}$ concentration ratio or without $PM_{2.5}$/$PM_{10}$ concentration ratio considered, we observed slightly higher estimates than those under the primary SDS events definition (Fig. S5). Results from sensitivity analyses, by changing the degree of freedom of spline functions, changing the adjustment of meteorological parameters (Fig. S6), using the data of different study periods (Fig. S7), and using the data of different study counties (Fig. S8), generally remained consistent with those from the main models for most mortality outcomes.

## Discussion

In this nationwide study, we comprehensively investigated the cause-specific mortality risks of short-term exposure to SDS events using the data of 1,495,724 deaths from 214 Chinese counties during the SDS period between 2013 and 2018. To the best of our knowledge, this study is the first to elucidate the mortality risks of SDS using a large sample size and a spectrum of mortality outcomes. Respiratory mortality significantly and substantially increased during SDS event days compared with clean days (8.90%; 95% CI: 4.96%, 12.98%). We identified a spectrum of SDS-sensitive health outcomes, including ischemic stroke mortality, intracerebral hemorrhagic stroke mortality, hypertensive heart disease mortality, myocardial infarction mortality, acute myocardial infarction mortality, acute ischemic heart disease mortality, chronic lower respiratory disease mortality, and COPD mortality. Added effects of SDS events were observed for mortality due to ischemic stroke, chronic lower respiratory disease, and COPD. Findings from this study provided scientific evidence to deepen the current understanding of SDS health effects and to plan interventions to protect the public against SDS.

Evidence exists regarding increased respiratory mortality risk during SDS events;[5,8,16,17] for example, one study conducted in Italy reported an increase of 22% (95% CI: 4.0%, 43.1%) in respiratory mortality during SDS events[8], another study in Korea found SDS-associated respiratory mortality risk increased 2.43% (95% CI: −3.30%, 8.50%)[17]. Although SDS events are consistently found to be linked with respiratory outcomes, it is hard to compare the quantitative effect estimates across studies, as studies used different definitions of SDS events and analytical techniques[6]. For cause-specific respiratory

**Table 1 | Official sand-dust weather records during the sand and dust storms period (1 February–31 May) from 2013 to 2018 in China**

| Year | Sand-dust weather record ID (start and end time) |
|---|---|
| 2013 | 201301 (24 February), 201302 (28 February–1 March), 201303 (5 March–6 March), 201304 (8 March–10 March), 201305 (11 March–12 March), 201306 (7 April), 201307 (17 April–18 April), 201308 (18 May) |
| 2014 | 201401 (19 March), 201402 (26 March–27 March), 201403 (2 April–3 April), 201404 (23 April–24 April), 201405 (29 April–1 May), 201406 (8 May–9 May), 201407 (22 May–25 May) |
| 2015 | 201501 (21 February–22 February), 201502 (2 March), 201503 (8 March), 201504 (14 March), 201505 (27 March–29 March), 201506 (31 March–1 April), 201507 (15 April), 201508 (27 April–29 April), 201509 (30 April), 201510 (5 May), 201511 (10 May), 201512 (17 May), 201513 (31 May) |
| 2016 | 201601 (18 February–19 February), 201602 (27 February), 201603 (3 March–4 March), 201604 (17 March), 201605 (31 March–2 April), 201606 (6 April), 201607 (15 April), 201608 (21 April–22 April), 201609 (30 April–1 May), 201610 (5 May–6 May), 201611 (10 May–12 May) |
| 2017 | 201702 (19 February–21 February), 201703 (12 March), 201704 (23 March), 201705 (17 April), 201706 (19 April), 201707 (3 May–7 May), 201708 (28 May–29 May) |
| 2018 | 201801 (8 February–9 February), 201802 (14 March–17 March), 201803 (18 March–20 March), 201804 (26 March–29 March), 201805 (1 April–3 April), 201806 (4 April–6 April), 201807 (9 April–10 April), 201808 (13 April–14 April), 201809 (16 April–17 April), 201810 (21 May–23 May), 201811 (24 May–26 May) |

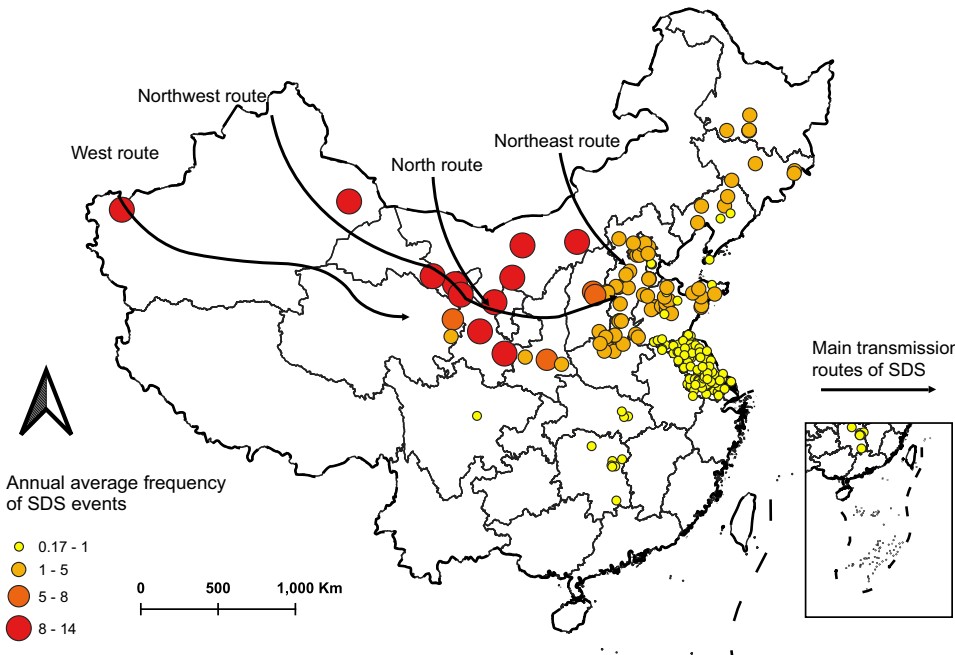

**Fig. 1 | Annual average frequency of Sand and Dust Storms (SDS) events in study county, 2013–2018.** Circles represent study counties. The colors of the circles represent the annual average frequency of SDS events for each county. The base map is the distribution of China's provinces, which was drawn based on the map data [Map Content Approval Number: GS (2022) No. 1873] from the official website of the Ministry of Civil Affairs of the People's Republic of China (http://xzqh.mca.gov.cn/map). Source data are provided as a Source Data file.

**Table 2 | Summary statistics for daily mortality in study counties during the Sand and Dust Storms (SDS) period (1 February–31 May) from 2013 to 2018**

| ICD-10 | Cause of mortality | Total deaths during the SDS period | Mean daily deaths during the SDS period (range) | Total deaths occurred on SDS event days |
|---|---|---|---|---|
| A00-Z99 | All (ALL) | 1,495,724 | 12.5 (0,116) | 19,082 |
| A00-R99 | Non-accidental (TOTAL) | 1,406,898 | 11.7 (0,116) | 17,923 |
| *Broad category* | | | | |
| I00-I99 | Diseases of the circulatory system (CIR) | 666,474 | 5.6 (0,110) | 9066 |
| J00-J99 | Diseases of the respiratory system (RES) | 173,632 | 1.6 (0,24) | 2156 |
| K00-K93 | Diseases of the digestive system (DSD) | 26,504 | 0.3 (0,6) | 321 |
| G00-G99 | Diseases of the nervous system (NSD) | 21,816 | 0.2 (0,9) | 185 |
| N00-N99 | Diseases of the genitourinary system (GSD) | 13,452 | 0.2 (0,6) | 181 |
| *Specific category* | | | | |
| I00-I99 | Diseases of the circulatory system (CIR) | | | |
| I20-I25 | Ischemic heart disease (IHD) | 268,305 | 2.5 (0,43) | 3896 |
| I20-I22, I24 | Acute ischemic heart disease (AIHD) | 152,585 | 1.5 (0,27) | 2518 |
| I21-I23 | Myocardial infarction (MI) | 151,335 | 1.5 (0,27) | 2459 |
| I21-I22 | Acute myocardial infarction (AMI) | 151,264 | 1.5 (0,27) | 2,456 |
| I25 | Chronic ischemic heart disease (CIHD) | 125,496 | 1.3 (0,36) | 1554 |
| I63 | Ischemic stroke (ISTR) | 114,376 | 1.1 (0,36) | 1487 |
| I60-I61 | Hemorrhagic stroke (HSTR) | 101,592 | 1.0 (0,34) | 1486 |
| I61 | Intracerebral hemorrhagic stroke (IHDSTR) | 94,827 | 0.9 (0,34) | 1400 |
| I11 | Hypertensive heart disease (HBP) | 41,640 | 0.5 (0,19) | 636 |
| **J00-J99** | **Diseases of the respiratory system (RES)** | | | |
| J40-J47 | Chronic lower respiratory disease (CLRI) | 124,744 | 1.2 (0,23) | 1571 |
| J41-J44 | Chronic obstructive pulmonary disease (COPD) | 119,944 | 1.1 (0,23) | 1514 |

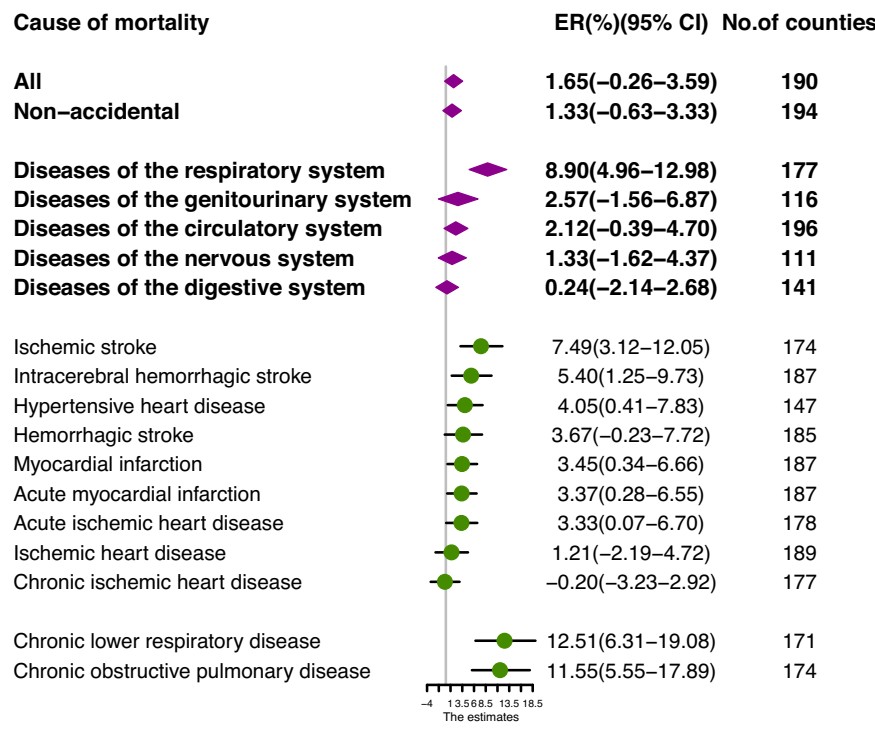

**Fig. 2 | Mortality risk associated with sand and dust storms events at lag0 day.** Estimates are shown for mortality due to broad categories of diseases (purple) and specific categories of diseases (green). Points represent the estimated excess risk (ER, %). Horizontal lines represent the 95% confidence interval (CI). The number of counties used to estimate the pooled estimates in the two-stage time series analysis is shown. Source data are provided as a Source Data file.

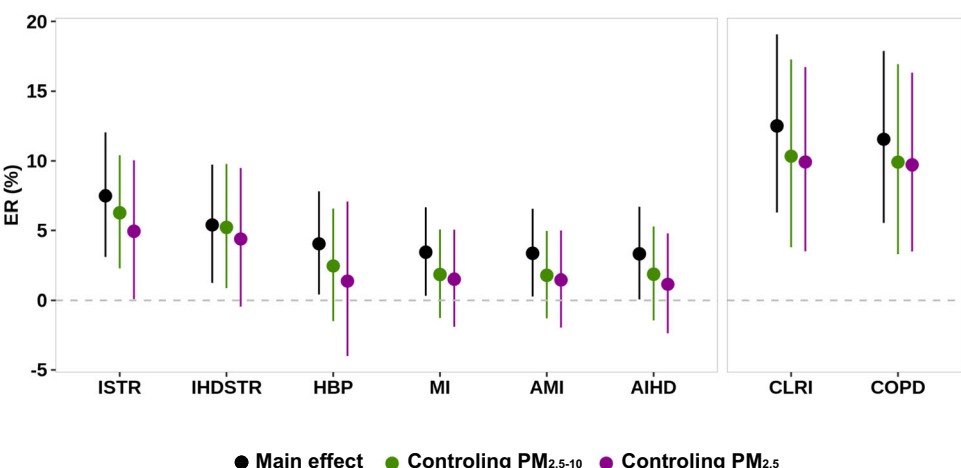

**Fig. 3 | Added effects of Sand and Dust Storms (SDS) events on mortality from SDS-sensitive health outcomes at lag0 day.** Results are shown for mortality risks associated with SDS from main analysis ("Main effect"), ("Controlling PM$_{2.5-10}$"), and ("Controlling PM$_{2.5}$"). ISTR: ischemic stroke mortality; IHDSTR: intracerebral hemorrhagic stroke mortality; HBP: hypertensive heart disease mortality; MI: myocardial infarction mortality; AMI: acute myocardial infarction mortality; AIHD: acute ischemic heart disease mortality; CLRI: chronic lower respiratory disease mortality; COPD: chronic obstructive pulmonary disease mortality. Points represent the estimated excess risk (ER, %). Vertical lines represent the 95% confidence interval (CI). The number of counties used to estimate the pooled estimates for each mortality outcome was the same among these three analyses. Specifically, for ISTR, IHDSTR, HBP, MI, AMI, AIHD, CLRI, and COPD, the corresponding numbers of counties were 174, 187, 147, 187, 187, 178, 171, and 174. Source data are provided as a Source Data file.

mortality outcomes, we found COPD mortality substantially increased (11.55%; 95% CI: 5.55%, 17.89%) during SDS, indicating COPD patients could be highly vulnerable to the adverse effects of SDS. This finding was consistent with previous relevant studies analyzing the effects of SDS on respiratory morbidity[18–20]; for example, a study estimated COPD hospitalization increased by 16% (95% CI: 8%, 24%) during SDS in a city of Southern Israel[19]. The elevated respiratory mortality risk associated with SDS may be due to considerable dust particles inhaled into the central airways[7,21]. Inhaling dust particles could physically harm the alveolar walls and bronchial epithelial cells[7]. The main components of dust particles, such as minerals including silicon dioxide and aluminum oxide, have been suggested to induce intense irritation and inflammation in the murine lung[22–25]. In addition, dust particles can carry microorganisms, such as bacteria, fungi, and viruses[6]. With the mice exposed in vivo to dust particles, Ichinose et al. found that these microbial antigens, such as lipopolysaccharide, adhered to the

particles' surfaces could cause an increase in pulmonary eosinophils[26,27]. COPD patients usually breathe with a large volume at a slow pace to overcome shortness of breath, resulting in a high concentration of dust particles in central airways, causing more severe irritation and inflammations and even leading to death[28].

Although the effect of SDS events on circulatory mortality in our study was not statistically significant (2.12%; 95% CI: −0.39%, 4.70%), our effect estimate was similar to the pooled estimate of SDS events' mortality risk (2.33%; 95% CI: 0.76%, 3.93%) as reported in a meta-analysis on the health effects of SDS in Asia[6]. Similarly, dust particles are associated with the onset of circulatory diseases[29,30]. The association of SDS events exposure with overall circulatory morbidity and mortality seemed pathologically plausible; for example, Cao et al.'s study showed an increase in circulating inflammatory cytokines and enzymes with rats' repeated exposure to fine dust particles[31]. Yet, little is known regarding how SDS exposure affects mortality due to cause-specific circulatory diseases. SDS exposure was significantly associated with mortality from several thrombotic diseases in this study, with ischemic stroke mortality having the leading risk (7.49%; 95% CI: 3.12%, 12.05%). Previous studies also showed a positive but nonsignificant association between SDS events and ischemic stroke using emergency visits and hospital admissions data[11,30]. We also observed that SDS events were associated with mortality from high blood pressure-related diseases, including intracerebral hemorrhagic stroke mortality (5.40%; 95% CI: 1.25%, 9.73%) and hypertensive heart disease mortality (4.05%; 95% CI: 0.41%, 7.83%). Yang et al. observed a 15% increase in primary intracerebral hemorrhagic stroke admissions associated with SDS event exposure at lag 3 day[11]. The systemic oxidative stress and inflammation induced by the inhalation of dust particles can cause vascular endothelial damage, increased platelet activity, and enhanced coagulation, thereby promoting thrombus formation[32,33]. There is also evidence that high concentrations of dust particles can cause an increase in heart rate and blood pressure[34,35], suggesting the plausibility of the associations of SDS events with stroke mortality and hypertensive heart disease mortality found in our study.

Our study first examined the effects of SDS events on mortality due to genitourinary, nervous, and digestive system diseases; we found positive associations between SDS events and these mortality outcomes, though not statistically significant. These were consistent with several epidemiological studies focusing on morbidity and experimental studies. For example, Herrera-Molina et al. found that exposure to SDS events was associated with an increased risk of hospitalizations from genitourinary diseases[36]. By collecting blood and urine samples from people affected by the dust storm, Badee-nezhad et al. measured biomarkers related to the central nervous system and found that $PM_{10}$ during SDS could cause neuron and astrocyte damage, leading to neuropsychiatric disorders[37]. Cao et al. found that repeated exposure to fine dust particles could cause pathological changes in the stomach of rats[31]. Our study results and previous findings provide evidence suggesting the potentially harmful impact of SDS events on mortality due to genitourinary, nervous, and digestive system diseases. More research is needed to investigate the effects of SDS on other diseases in addition to cardiorespiratory diseases.

The differences of health effects between SDS events and conventional PM pollution events (that is, $PM_{2.5}$ pollution events that happen on non-SDS event days) were confirmed by our results. For ischemic stroke, intracerebral hemorrhagic stroke, hypertensive heart disease, chronic lower respiratory disease, and COPD, SDS events could trigger a more serious impact than conventional PM pollution events. This may be due to the differences in the constitutes of PM between SDS events and conventional PM pollution events. Sources of PM during SDS include both anthropogenic and natural sources, which are more complicated[26]. He et al. reported that $PM_{2.5}$ during SDS had a greater exacerbating effect on the lung eosinophilia of mice than $PM_{2.5}$

in hazy weather[38]. This could result from the increased bioreactivity of $PM_{2.5}$ during SDS. Ho et al. found that significant amounts of suspended dust particles during SDS provided platforms to intermix with chemicals on their surfaces[39]. These reactions may be an unrecognized source of toxic compositions, enhancing the $PM_{2.5}$ toxicity during SDS. Furthermore, fine dust particles can be elevated into the troposphere and travel thousands of kilometers[40], absorbing airborne pollutants from anthropogenic sources in industrial areas, microorganisms, and potential allergens, such as pollens, and increasing the adverse health effects of $PM_{2.5}$ during SDS[6,13].

When accounting for $PM_{2.5}$ and $PM_{2.5\text{-}10}$, we still observed added effects of SDS events on mortality from ischemic stroke, chronic lower respiratory disease, and COPD. Similarly, Sun et al. observed that heavy $PM_{2.5}$ pollution events, defined as daily average $PM_{2.5}$ concentration ≥75 μg/m³ for at least 3 days, had added effects on the circulatory (0.96%; 95% CI: 0.37%, 1.55%) and respiratory (0.55%; 95% CI: −0.52%, 1.63%) mortality[15]. SDS events have coincided with high concentration levels of $PM_{2.5\text{-}10}$ and $PM_{2.5}$[4]. Experimental studies found that exposure to sustained high $PM_{2.5}$ concentrations could cause severe damage to multiple organs in mice, including cardiac fibrosis and myocardial hypertrophy[41,42]. In addition, SDS may evoke worry and stress which have been shown to contribute to health symptoms of all body systems, including vasoconstriction and increased blood pressure at a cardiovascular level[43–46].

This study has some limitations. First, exposure misclassification is possible as there was no universal definition for SDS events[47]. However, we considered the official sand-dust weather records, the $PM_{10}$ concentration, and $PM_{2.5}/PM_{10}$ concentration ratio in identifying SDS events in this study; we also used several alternative definitions for SDS events in sensitivity analysis, with the results pretty robust to the primary results. Second, the limited accessibility of mortality data hampered us to include more comprehensive and even national regions in China. However, to the best of our knowledge, this study is the largest epidemiological study to investigate the mortality risks associated with SDS. Third, we found significant added effects of SDS on cardiorespiratory mortality when accounting for PM exposures. Still, this study cannot distinguish natural and anthropogenic sources of PM, which is also not the scope of this study. More studies are needed to explore the independent health effects of SDS with natural and anthropogenic PM accounted for.

Findings from this study can provide implications for policymakers and the public. First, given the broad and severe SDS health impact, it is necessary to establish air quality guidelines and standards for SDS. Second, the health departments should allocate medical resources, especially for people in need, before SDS events. Finally, the public should be educated and informed about the potential health risks of SDS and adequate protective measures.

To summarize, this nationwide multicenter study showed that short-term exposure to SDS events is linked to increased mortality from many causes, particularly respiratory disease. Public health policy against SDS should be implemented, as SDS presents adverse health risks in addition to conventional PM pollution.

## Methods
This study complies with all relevant ethical regulations and was approved by the National Institute of Environmental Health, the Chinese Center for Disease Control and Prevention.

### Study design
We first collected data for counties frequently affected by SDS; the distribution of study counties was designed to cover the primary SDS transmission routes of China, with a good representation of the heterogeneity in exposure levels to SDS events (Fig. 1). We then performed a two-stage time series analysis using the daily data from 2013 to 2018 for 214 Chinese counties. Further, we investigated the added

effects of SDS events by controlling PM$_{2.5-10}$ and PM$_{2.5}$ in the models, respectively. Fig. S9 shows a diagram of our study design.

## Study sites and mortality data
We finally included 214 counties (Fig. 1), with the flowchart of selecting counties shown in Fig. S10. Daily mortality data were obtained from China's Disease Surveillance Points System of the Chinese Center for Disease Control and Prevention. We chose mortality outcomes that have been usually examined in previous epidemiologic studies on the health effects of SDS and PM pollution. We analyzed 18 mortality outcomes, including mortality due to 7 broad category causes and 11 specific category causes, based on the 10th version International Classification of Diseases (Table 1).

## Air pollution and meteorological data
Daily county-specific concentrations of air pollutants, including PM$_{10}$ and PM$_{2.5}$, were obtained from hourly data reported by China's National Air Pollution Monitoring System. For each county, concentrations of air pollutants were calculated using the available daily average data from all fixed monitoring sites located within the county. We calculated PM$_{2.5-10}$ concentration by subtracting PM$_{2.5}$ concentration from PM$_{10}$ concentration for each county[48]. Meteorological data, including temperature and relative humidity, were obtained from the ERA5-land reanalysis dataset released by European Centre for Medium-Range Weather Forecasts (https://cds.climate.copernicus.eu/cdsapp#!/dataset/reanalysis-era5-land?tab=overview). We extracted hourly data based on the geographical coordinates of county central point and calculated the daily measures for each county.

## SDS events definition
We collected the official sand-dust weather records for our study counties from China's National Meteorological Center (Table 1). Since the official sand-dust weather is recorded at the province level, we consider PM$_{10}$ concentration and PM$_{2.5}$/PM$_{10}$ concentration ratio in the SDS events definition. Specifically, for each study county, an SDS event was defined as a day when: (1) there was an official sand–dust weather record on the day; (2) the daily concentration of PM$_{10}$ was >50 μg/m$^3$ [47,49,50], which was the lowest threshold observed in Huffman et al.'s classification of PM$_{10}$ during SDS;[47,49] (3) the daily PM$_{2.5}$/PM$_{10}$ concentration ratio was <0.4 [18]. The PM$_{2.5}$/PM$_{10}$ concentration ratio is an important indicator to distinguish sand–dust weather from non-sand–dust weather, as the low ratio is often associated with overwhelming contribution from long-distance transport dust particles[51].

We also considered three alternative definitions of SDS events by using two different thresholds in the PM$_{2.5}$/PM$_{10}$ concentration ratio (0.35 and 0.45) and excluding the PM$_{2.5}$/PM$_{10}$ concentration ratio in the SDS events definition. More details of the SDS events definition are provided in the Supplementary Methods.

## Statistical analysis
Two-stage time series analysis was applied to estimate the associations between short-term exposure to SDS events and a spectrum of mortality outcomes. The primary analysis was conducted using data for the SDS period (1 February–31 May), which had a high frequency of SDS events from 2013 and 2018 (Fig. S1). In the first stage, we fit a generalized linear model (GLM) with quasi-Poisson distribution to assess the effects of SDS on mortality from a spectrum of causes for each county. This county-level analysis based on our mortality data has been shown feasible enough to ensure statistical power by previous researches[52–57] and our team's researches[15,58–60]. The model was fit for each mortality outcome separately. The model equation is as follows:

$$\log E(Y_t) = \text{Intercept} + \beta Z_t + ns(\text{Time}_t, df) \\ + ns(\text{Temp}_t, df) + ns(\text{RH}_t, df) + \text{Dow}_t \quad (1)$$

where $Y_t$ is the number of deaths on day $t$; $Z_t$ represents the exposure on day $t$; to distinguish the effect of PM pollution on non-SDS days, $Z_t$ was a categorical variable with "1" for an identified SDS event day, "2" for a non-SDS event day with PM$_{2.5}$ pollution (that is, daily PM$_{2.5}$ concentration ≥75 μg/m$^3$ on a non-SDS day), and "0" for a clean day (that is, neither SDS event day nor PM$_{2.5}$ pollution day); $\beta$ is a vector of coefficients with the length of two. We used the natural spline functions ("$ns$") in controlling the confounding of long-time trend, daily mean temperature ("Temp$_t$"), and daily relative humidity ("RH$_t$"), with the degrees of freedom ($df$) of 2 (per SDS period), 3, and 3, respectively. Dow$_t$ is an indicator variable denoting the day of the week. Based on the estimated $\beta$ coefficients, we calculated the effects of SDS events on mortality as the observed mortality on SDS event days compared with clean days; we also reported the mortality risks on non-SDS event days with PM$_{2.5}$ pollution.

We used a random-effects meta-analysis in the second stage to obtain the pooled effect estimates for study counties. The following equation calculated ER for mortality associated with SDS events:

$$\text{ER} = \left[\exp(\beta_{meta}) - 1\right] \times 100\% \quad (2)$$

where $\beta_{meta}$ denotes the pooled effect estimates from meta-analysis.

Moreover, we conducted stratified analyses by sex (male and female) and age (<75 and ≥75 years) for all-cause, non-accidental, circulatory, and respiratory mortality. The differences in estimates between different subgroups were evaluated using Z tests.

Since SDS events are typically PM pollution, we further investigate whether SDS events exposure had added effects on mortality by adjusting for PM$_{2.5-10}$ and PM$_{2.5}$ in the GLM models. In addition, to investigate the potential delayed effects of SDS events exposure, we performed a lagged analysis by fitting the same model separately for three single-day lagged data (lag 1, 2, and 3). For example, analysis at lag 1 estimates the impact on mortality on day t ($Y_t$) associated with exposure to the previous day (i.e., $Z_{t-1}$). In the lagged analysis, we used data for the same lagged day for Temp$_t$ and RH$_t$.

## Sensitivity analyses
We conducted a series of sensitivity analyses to assess the robustness of the estimated associations between SDS events and many mortality outcomes in our primary analysis. First, we examined the mortality risk of SDS events under different SDS definitions. Second, we changed the df for the time trend variable ($df$ = 3) and used two different df ($df$ = 4, 5) for meteorological parameters in the spline functions. Third, instead of the daily mean temperature and relative humidity, we used the 21-day moving average of temperature and the 7-day moving average of relative humidity to fully adjust for the confounding of meteorological conditions[61]. Fourth, we refit the GLM model in the first stage using the whole year's data. Fifth, since the daily county-level death counts were pretty small for certain mortality outcomes, we conducted sensitivity analyses only on study counties with daily death counts exceeding one during the SDS periods. This approach allowed us to examine the potential uncertainty introduced by these low counts.

## Reporting summary
Further information on research design is available in the Nature Portfolio Reporting Summary linked to this article.

# Data availability
The data generated in this study are available under restricted access for the identifiable nature of the data and data management requirements. Access can be obtained by contacting the corresponding author (litiantian@nieh.chinacdc.cn) and will be answered within 12 weeks. The data can be used through collaborative research with authors. The exposure data for air pollution in this study was available from China's National Air Pollution Monitoring System (http://www.

cnemc.cn) upon request. The exposure data for meteorological data in this study were downloaded from the ERA5-land reanalysis dataset released by European Centre for Medium-Range Weather Forecasts (https://cds.climate.copernicus.eu/cdsapp#!/dataset/reanalysis-era5-land?tab=overview). Official sand–dust weather records were collected from the Sand-dust Weather Almanac compiled by the China Meteorological Administration, a book published by the Meteorological Publishing House. The electronic version of this book can be downloaded from the China Knowledge Network (https://www.cnki.net). Source data are provided with this paper.

## Code availability

Code used in this study is available online at (https://github.com/sunshineann/SDS_mortality_NatureComm).

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

## Acknowledgements

This study was supported by the National Natural Science Foundation of China (92143202, T.L.; 82241051, T.L.).

## Author contributions

T.L. conceived the study. C.C. processed the mortality data. M.Y., H.D. and Y.L. offered the methodology guiding. C.Z. performed the data analysis. C.Z. prepared tables, and figures and drafted paper. M.Y. assisted with the interpretation of results. T.L., M.Y. and J.B. reviewed and edited the paper. All authors contributed to the manuscript. All authors have given approval to the final version of the manuscript.

## Competing interests

The authors declare no competing interests.
