## [Peer Review File · Nature Communications]

Mortality risks from a spectrum of causes associated with sand and dust storms in ChinaReviewers' comments:

Reviewer #1 (Remarks to the Author):

Thank you for inviting me to peer review this manuscript. This interesting, original study employs a multicentre, nationwide, time series design to quantify the excess risk of mortality posed by sand and dust storms (SDS) in China. The findings are based on almost 1.5 million deaths over a period of 6 years, in 214 Chinese counties. Sand and dust storms are associated with 8.9% increase in mortality due to respiratory disease, that was mainly driven by COPD, and with a non-significant trend over increased mortality due to circulatory disease.

Major comments:

- The authors should state how many deaths coincided with the SDS events, as this will help the reader interpret the results.
- The authors should justify the selection of the “broad categories” and “specific categories” of diseases that were selected for evaluation.
- The investigators label together ICD-10 codes for bronchitis [acute or chronic], COPD, asthma and bronchiectasis as “chronic lower respiratory diseases” and suggest these are the leading cause of excess mortality due to SDS. However, COPD was responsible for >95% of the deaths that occurred due to “chronic lower respiratory diseases” and the study is not adequately powered to assess for excess asthma or bronchiectasis related mortality. Therefore, the authors should focus on COPD, rather than “chronic lower respiratory diseases”.
- In their analysis authors considered days with high PM2.5 concentration that were not associated with SDS. They should report on the excess mortality during these days and whether it was comparable with the SDS days [and with SDS days with comparable PM2.5].
- Based on their delayed effect analyses, could the authors comment on how long after the SDS is the mortality significantly raised?
- The authors need to describe in more detail how the lag0, lag1 etc variables were defined.
- It seems to me that the authors describe in their main analysis the excess mortality due to SDS at the day of the SDS. Is there a way of quantifying the overall impact of an SDS on mortality overtime?
- The discussion of this paper is weak. The potential mechanisms proposed for the various epidemiological findings are speculative and perhaps beyond the scope of this manuscript.

Minor comments:

- Authors should limit the use of acronyms. At present, they have too many and it is challenging for the reader to follow.

Reviewer #2 (Remarks to the Author):

This manuscript evaluated the cause-specific mortality risks associated with sand and storms in China. The strength is the use of a multicenter dataset comprising of 214 counties. This topic is not novel, given that many previous studies worldwide had examined these associations, albeit the number of death causes previously evaluated are limited.

Some specific concerns are listed below.

1. Line 69, why the number of counties used for the pool estimates of each death cause was >100? Please give more details.
2. Abbreviations in the results should be defined when they are firstly presented.
3. Line 99, a total of 1,495,724 deaths were used. It is unclear on whether they occurred during the SDS event periods? If they covered the whole year, this number is useless as only the SDS event period was explored.
4. Line 250, "all analyses were restricted to the period with a high frequency of SDS events during the study", why only February to May was included? From Figure S3, some other seasons also had SES events.
5. More importantly, the first-stage analysis was conducted at the county level. However, as showed in Table 1, the daily cases for many specific causes of deaths were quite few (equal or lower than 1 case), which would greatly attenuate the statistical power in an ecological time-series analysis.
6. Line 262, only 2 df per year for time trends may be not sufficient.
7. Line 264, the lagged effects of weather conditions should be also controlled. 3 df for temperature is also not enough.
8. Line 278, "Series of sensitivity analyses" should be also specified.

Reviewer #3 (Remarks to the Author):

In this paper the authors aim to produce novel evidence on the association between cause-specific mortality and sand and dust storms in China. The paper has the merit to collect a substantial amount of data from multiple areas of China. However, there are several flaws in the statement of the objectives, the details of the statistical analysis, the wording of many sentences (sometimes truncated), and the interpretation of the results.

First, the paper mentions a spectrum of causes in the title, and indeed evaluates many of them, but little is reported in the interpretation of results in terms of cause-specific effect estimates. Adverse effects of dust on CVD and RESP mortality are largely known, and it is not clear how the paper contributes on the topic.

Second, some analyses (such as effect modification by age and sex) are not helpful in capturing the main message of the paper, but rather add noise to the overall picture, and are not well motivated.

Concerning the methods:

1. it is not clear why the authors define a 3 level exposure 0-1-2 variable rather than 2 levels 0-1 (what is the purpose of having a level 2?)

2. the definition of SDS days is totally driven by PM concentrations, and the argument that when PM exceeds the thresholds and the ratio is above 0.4 it should be a dust day is only speculative, not supported by any data. Why didn't the authors use external sources of info, such as atmospheric models?

3. some choices (number of DF for time trends or temperature, lag terms for temperature) are questionable and not adequately motivated (despite the sensitivity analyses)

4. the presumed "contribution" of PM_{2.5} and PM_{2.5-10} is purely speculative: what the authors estimate is the effect of SDS 0/1 with or without adjustment for PM. The difference between such estimates does not say anything on the role of PM on cause-specific mortality during SDS days,

As a consequence, the interpretation of the results looks overstated, and the general picture is chaotic.

1 **Responses to reviewers' comments for "Mortality risks from a spectrum of**
2 **causes associated with sand and dust storms in China"**

3 Thanks, editor, for your time processing our manuscript and the reviewers' professional
4 suggestions. We have revised the relevant parts accordingly and responded to the
5 reviewers' comments. We hope you will be more satisfied with this manuscript once
6 we have addressed these issues and concerns in the comments.

7
8 **Responses to Reviewer #1:**

9 Thank you for inviting me to peer review this manuscript. This interesting, original
10 study employs a multicentre, nationwide, time series design to quantify the excess risk
11 of mortality posed by sand and dust storms (SDS) in China. The findings are based on
12 almost 1.5 million deaths over a period of 6 years, in 214 Chinese counties. Sand and
13 dust storms are associated with 8.9% increase in mortality due to respiratory disease,
14 that was mainly driven by COPD, and with a non-significant trend over increased
15 mortality due to circulatory disease.

16 **Response:**

17 Thanks for your time and efforts in reviewing our manuscript. We have thoroughly
18 revised our manuscript and addressed these issues raised from your comments, with a
19 point-by-point response as provided below.

20
21 Major comments:

22 - The authors should state how many deaths coincided with the SDS events, as this will
23 help the reader interpret the results.

24 **Response:**

25 Thanks for your suggestion. We have added the number of deaths occurred on the SDS
26 event days in Table 2 and corresponding text.

27
28 **Relevant text in revised Results section:**

29 “A total of 1,495,724 deaths occurred during the SDS period, and 19,082 deaths in the
 30 identified SDS event days (Table 2).” (Lines 60-61)

31

32 **Table 2. Summary statistics for daily mortality in study counties during the SDS**
 33 **period (1 February-31 May) from 2013 to 2018**

34

ICD-10	Cause of mortality	Total deaths during the SDS period	Mean daily deaths during the SDS period (range)	Total deaths occurred on SDS event days
A00-Z99	All (ALL)	1,495,724	12.5 (0,116)	19,082
A00-R99	Non-accidental (TOTAL)	1,406,898	11.7 (0,116)	17,923
Broad category				
I00-I99	Diseases of the circulatory system (CIR)	666,474	5.6 (0,110)	9,066
J00-J99	Diseases of the respiratory system (RES)	173,632	1.6 (0,24)	2,156
K00-K93	Diseases of the digestive system (DSD)	26,504	0.3 (0,6)	321
G00-G99	Diseases of the nervous system (NSD)	21,816	0.2 (0,9)	185
N00-N99	Diseases of the genitourinary system (GSD)	13,452	0.2 (0,6)	181
Specific category				
I00-I99	Diseases of the circulatory system (CIR)			
I20-I25	Ischemic heart disease (IHD)	268,305	2.5 (0,43)	3,896
I20-I22, I24	Acute ischemic heart disease (AIHD)	152,585	1.5 (0,27)	2,518
I21-I23	Myocardial infarction (MI)	151,335	1.5 (0,27)	2,459
I21-I22	Acute Myocardial infarction (AMI)	151,264	1.5 (0,27)	2,456
I25	Chronic ischemic heart disease (CIHD)	125,496	1.3 (0,36)	1,554
I63	Ischemic stroke (ISTR)	114,376	1.1 (0,36)	1,487
I60-I61	Hemorrhagic stroke (HSTR)	101,592	1.0 (0,34)	1,486
I61	Intracerebral hemorrhagic stroke (IHDSTR)	94,827	0.9 (0,34)	1,400
I10-I15	Hypertensive heart disease (HBP)	41,640	0.5 (0,19)	636
J00-J99	Diseases of the respiratory system (RES)			
J40-J47	Chronic lower respiratory disease (CLRI)	124,744	1.2 (0,23)	1,571
J41-J44	Chronic obstructive pulmonary disease (COPD)	119,944	1.1 (0,23)	1,514

- The authors should justify the selection of the “broad categories” and “specific categories” of diseases that were selected for evaluation.

Response:

Thanks for your suggestion. We selected the “broad categories” and “specific categories” of diseases as those were often examined in relevant studies on the health effects of particulate matter and SDS. We have added more details to justify the selection of the broad and specific categories in the revised manuscript.

Relevant text in revised Methods section:

“Daily mortality data were obtained from China’s Disease Surveillance Points System of the Chinese Center for Disease Control and Prevention. We chose mortality outcomes that have been usually examined in previous epidemiologic studies on the health effect of SDS and PM pollution.” (Lines 265-268)

- The investigators label together ICD-10 codes for bronchitis [acute or chronic], COPD, asthma and bronchiectasis as “chronic lower respiratory diseases” and suggest these are the leading cause of excess mortality due to SDS. However, COPD was responsible for >95% of the deaths that occurred due to “chronic lower respiratory diseases” and the study is not adequately powered to assess for excess asthma or bronchiectasis related mortality. Therefore, the authors should focus on COPD, rather than “chronic lower respiratory diseases”.

Response:

We appreciate your professional suggestion. We focused our discussion on COPD mortality rather than “chronic lower respiratory diseases”.

Relevant text in revised Discussion section:

“For cause-specific respiratory mortality outcomes, we found COPD mortality substantially increased (11.55%; 95% CI: 5.55%, 17.89%) during SDS, indicating COPD patients could be highly vulnerable to SDS’s adverse effects. This finding was consistent with previous relevant studies analyzing the effects of SDS on respiratory

morbidity¹⁻³; for example, a study estimated COPD hospitalization increased by 16% (95% CI: 8%, 24%) during SDS in a city of Southern Israel².” (Lines 133-139)

References for responses:

1. Tam, W. W. S., Wong, T. W., Wong, A. H. S. & Hui, D. S. C. Effect of dust storm events on daily emergency admissions for respiratory diseases. *Respirology* **17**, 143–148 (2012).
2. Vodonos, A. *et al.* The impact of desert dust exposures on hospitalizations due to exacerbation of chronic obstructive pulmonary disease. *Air Qual. Atmos. Health* **7**, 433–439 (2014).
3. Lorentzou, C. *et al.* Extreme desert dust storms and COPD morbidity on the island of Crete. *Int. J. Chron. Obstruct. Pulmon. Dis.* **14**, 1763–1768 (2019).

- In their analysis authors considered days with high PM_{2.5} concentration that were not associated with SDS. They should report on the excess mortality during these days and whether it was comparable with the SDS days [and with SDS days with comparable PM_{2.5}].

Response:

Thanks for your suggestion. We have added the estimates of mortality risks during non-SDS event days with high PM_{2.5} concentration in the revised manuscript. We added the corresponding discussion about the differences of health effects between SDS event days and non-SDS event days with PM_{2.5} pollution. We also very appreciate your professional suggestion about the report on the excess mortality during SDS days with comparable PM_{2.5}. The current health effect evidence of SDS comes from studies that assessed the health risks of SDS events, comparing the occurrence of health outcomes during SDS event days and non-SDS event days, and from studies that considered SDS events as an effect modifier for the health effects of any given PM fraction^{1,2}. For instance, in 2020, Hashizume M. and colleagues published the first meta-analysis about the health effect of Asian SDS events based on 21 studies in *Environmental Health Perspectives*³. Mallone S. *et al* focused on investigating the mortality risks associated

with PM_{2.5}, PM_{2.5-10}, PM₁₀ during SDS event days and non-SDS event days in Rome, Italy⁴. Your suggestion would help to understand the difference of heavy PM_{2.5} pollution impacts during different periods (SDS event days, non-SDS event days), which is a valuable point in terms of SDS related studies. However, in this study, our main purpose was to investigate the mortality risks of SDS events, providing information on the magnitudes of health effect, SDS-sensitive health outcomes, temporal distribution patterns, and the added effects of SDS events. The difference of PM_{2.5} pollution impacts during different periods (SDS event days, non-SDS event days) is not essential in our study. Thus, our results did not include the estimates of mortality risks during SDS event days with comparable PM_{2.5}. Whereas we are very interested in exploring it in our future study. We thank you again for the great ideas your suggestions have given us.

Relevant text in revised Results section:

“Compared to clean days, we observed a more enhanced risk of mortality for ischemic stroke, intracerebral hemorrhagic stroke, hypertensive heart disease, chronic lower respiratory disease, and COPD on SDS event days than non-SDS event days with PM_{2.5} pollution (Fig. S3).” (Lines 85-88)

Fig. S3. Excess risk (ER, %) and 95% confidence intervals for mortality associated with sand and dust storms (SDS) event day (left) or non-SDS event day with PM_{2.5} pollution (right). PM_{2.5} pollution represents that daily PM_{2.5} concentration ≥ 75 $\mu\text{g}/\text{m}^3$. Red represents that the ER differed significantly from 0% ($P < 0.05$). The Mortality of broad causes results are shown in the top two panels, and mortality of specific causes results in the bottom two panels.

Relevant text in revised Discussion section:

“The differences of health effects between SDS events and conventional PM pollution events (that is, PM_{2.5} pollution events which happen on non-SDS event days) were confirmed by our results. For ischemic stroke, intracerebral hemorrhagic stroke, hypertensive heart disease, chronic lower respiratory disease and COPD, SDS events could trigger more serious impact than conventional PM pollution events. This may be due to the differences in constituents of PM between SDS events and conventional PM pollution events. Sources of PM during SDS include both anthropogenic and natural sources, which is more complicated⁵. He et al.⁶ reported that PM_{2.5} during SDS had greater exacerbating effect on the lung eosinophilia of mice than PM_{2.5} in hazy weather. This could result from the increase bioreactivity of PM_{2.5} during SDS. Ho et al.⁷ found that significant amounts of suspended dust particles during SDS provided platforms to

intermix with chemicals on their surfaces. And these reactions may be an unrecognized source of toxic compositions, enhancing the PM_{2.5} toxicity during SDS. Furthermore, fine dust particles can be elevated into the troposphere and travel thousands of kilometers⁸, absorbing airborne pollutants from anthropogenic sources in industrial areas, microorganisms, and potential allergens, such as pollens, and increasing the adverse health effects of PM_{2.5} during SDS^{3,9}.” (Lines 196-212)

References for responses:

1. WHO. WHO global air quality guidelines: particulate matter (PM_{2.5} and PM₁₀), ozone, nitrogen dioxide, sulfur dioxide and carbon monoxide. (2021).
2. Lwin, K. S. *et al.* Effects of desert dust and sandstorms on human health: A scoping review. *GeoHealth* **7**, e2022GH000728 (2023).
3. Hashizume, M. *et al.* Health effects of Asian dust: A systematic review and meta-analysis. *Environ. Health Persp.* **128**, 066001 (2020).
4. Mallone, S. *et al.* Saharan dust and associations between particulate matter and daily mortality in Rome, Italy. *Environ. Health Persp.* **119**, 1409–1414 (2011).
5. Li G. *et al.* Resurgence of sandstorms complicates China's air pollution situation. *Environ. Sci. Technol.* **55**, 11467-11469 (2021).
6. He M. *et al.* Differences in allergic inflammatory responses between urban PM_{2.5} and fine particle derived from desert-dust in murine lungs. *Toxicol. Appl. Pharm.* **297**, 41-55 (2016).
7. Ho K. F. *et al.* Contributions of local pollution emissions to particle bioreactivity in downwind cities in China during Asian dust periods. *Environ. Pollut.* **245**, 675-683 (2019).
8. Middleton N., Kang U. Sand and dust storms: Impact mitigation. *Sustainability* **9**, 1053 (2017).
9. Wu Y., Wen B., Li S., Guo Y. Sand and dust storms in Asia: A call for global cooperation on climate change. *Lancet Planet. Health* **5**, e329-e330 (2021).

- Based on their delayed effect analyses, could the authors comment on how long after the SDS is the mortality significantly raised?

Response:

We observed significantly increased mortality risk at the lag 0 day for SDS events; relevant texts were added in the revise manuscript.

Relevant text in revised Results section:

“The results of delayed effect of SDS are provided in the Supplement (Fig. S2). And the highest and significant mortality risks from most diseases were observed at day lag0. Similar effect estimates were also observed at lag 1 day, while more negligible effects or no associations at the next 2–3 days (Fig. S2).” (Lines 80-83)

- The authors need to describe in more detail how the lag0, lag1 etc variables were defined.

Response:

Thanks for your suggestion. We have added more details on the definitions of lagged variables (i.e., lag0, lag1) in the revised manuscript. We hope this would be clearer to readers.

Relevant text in revised Methods section:

“In addition, to investigate the potential delayed effects of SDS events exposure, we performed lagged analysis by fitting the same model separately for three single-day lagged data (lag 1, 2, and 3). For example, analysis at lag 1 estimates the impact on mortality on day t (Y_t) associated with exposure to the previous day (i.e., Z_{t-1}). In the lagged analysis, we used data for the same lagged day for $Temp_t$ and RH_t .” (Lines 341-345)

- It seems to me that the authors describe in their main analysis the excess mortality due to SDS at the day of the SDS. Is there a way of quantifying the overall impact of an SDS on mortality overtime?

Response:

Thanks for your valuable suggestion. For calculating excess mortality, the overall impact you mentioned is indeed more appropriate. However, a large weight of our study is currently carving out the mortality risks of SDS events, providing information on the magnitudes of health effect, SDS-sensitive health outcomes, temporal distribution patterns, and the added effects of SDS events. We expect that the findings of this study will provide scientific evidence to deepen the current understanding of SDS's health effects and to plan interventions to protect the public against SDS.

In view of the main objective of our article, and taking into account the comments of the reviewers 3 in this piece, we thought that the calculation of excess mortality is not essential in order to make the topic of our article clear. Therefore, we have removed this section in this study.

- The discussion of this paper is weak. The potential mechanisms proposed for the various epidemiological findings are speculative and perhaps beyond the scope of this manuscript.

Response:

Thanks for identifying this issue. We have thoroughly revised our discussion. Since this study aims to investigate the short-term association between SDS and mortality outcomes, we comprehensively discussed whether the observed associations could be biologically plausible. In addition, we discussed the potential reasons for the differences of health effects between SDS events and conventional PM pollution events (that is, PM_{2.5} pollution events which happen on non-SDS event days). We also added the discussion for added effects of SDS events on mortality when accounting for PM in the revised manuscript.

Relevant text in revised Discussion section:

“In this nationwide study, we comprehensively investigated the cause-specific mortality risks of short-term exposure to SDS events using the data of 1,495,724 deaths from 214

Chinese counties during the SDS period between 2013 and 2018. To the best of our knowledge, this study is the first to elucidate the mortality risks of SDS using a large sample size and a spectrum of mortality outcomes. Respiratory mortality significantly and substantially increased during SDS event days compared with clean days (8.90%; 95% CI: 4.96%, 12.98%). We identified a spectrum of SDS-sensitive health outcomes, including ischemic stroke mortality, intracerebral hemorrhagic stroke mortality, hypertensive heart disease mortality, myocardial infarction mortality, acute myocardial infarction mortality, acute ischemic heart disease mortality, chronic lower respiratory disease mortality, and COPD mortality. Added effects of SDS events were observed for mortality due to ischemic stroke, chronic lower respiratory disease, and COPD. Findings from this study provided scientific evidence to deepen the current understanding of SDS's health effects and to plan interventions to protect the public against SDS.

Evidence exists regarding increased respiratory mortality risk during SDS events^{1, 2, 3, 4}; for example, one study conducted in Italy reported an increase of 22% (95% CI: 4.0%, 43.1%) in respiratory mortality during SDS events³, another study in Korea found SDS-associated respiratory mortality risk increased 2.43% (95% CI: -3.30%, 8.50%)⁴. Although SDS events are consistently found to be linked with respiratory outcomes, it is hard to compare the quantitative effect estimates across studies, as studies used different definitions of SDS events and analytical techniques⁵. For cause-specific respiratory mortality outcomes, we found COPD mortality substantially increased (11.55%; 95% CI: 5.55%, 17.89%) during SDS, indicating COPD patients could be highly vulnerable to SDS's adverse effects. This finding was consistent with previous relevant studies analyzing the effects of SDS on respiratory morbidity^{6, 7, 8}; for example, a study estimated COPD hospitalization increased by 16% (95% CI: 8%, 24%) during SDS in a city of Southern Israel⁷. The elevated respiratory mortality risk associated with SDS may be due to considerable dust particles inhaled into the central airways^{9, 10}. Inhaling dust particles could physically harm the alveolar walls and bronchial epithelial cells⁹. The main components of dust particles, such as minerals

including silicon dioxide and aluminum oxide, have been suggested to induce intense irritation and inflammation in the murine lung^{11, 12, 13, 14}. In addition, dust particles can carry microorganisms, such as bacteria, fungi, and viruses⁵. With the mice exposed in vivo to dust particles, Ichinose et al. found that these microbial antigens, such as lipopolysaccharide, adhered to the particles' surfaces could cause an increase in pulmonary eosinophils^{15, 16}. COPD patients usually breathe with a large volume at a slow pace to overcome shortness of breath, resulting in a high concentration of dust particles in central airways, causing more severe irritation and inflammations and even leading to death¹⁷.

Although the effect of SDS events on circulatory mortality in our study was not statistically significant (2.12%; 95% CI: -0.39%, 4.70%), our effect estimate was similar to the pooled estimate of SDS events' mortality risk (2.33%; 95% CI: 0.76%, 3.93%) as reported in a meta-analysis on the health effects of SDS in Asia⁵. Similarly, dust particles are associated with the onset of circulatory diseases^{18, 19}. The association of SDS events exposure with overall circulatory morbidity and mortality seemed pathologically plausible; for example, Cao et al.'s study²⁰ showed an increase in circulating inflammatory cytokines and enzymes with rats' repeated exposure to fine dust particles. Yet, little is known regarding how SDS exposure affects mortality due to cause-specific circulatory diseases. SDS exposure was significantly associated with mortality from several thrombotic diseases in this study, with ischemic stroke mortality having the leading risk (7.49%; 95% CI: 3.12%, 12.05%). Previous studies also showed a positive but nonsignificant association between SDS events and ischemic stroke using emergency visits and hospital admissions data^{19, 21}. We also observed that SDS events were associated with mortality from high blood pressure-related diseases, including intracerebral hemorrhagic stroke mortality (5.40%; 95% CI: 1.25%, 9.73%) and hypertensive heart disease mortality (4.05%; 95% CI: 0.41%, 7.83%). Yang et al.²¹ observed a 15% increase in primary intracerebral hemorrhagic stroke admissions associated with SDS event exposure at lag 3. The systemic oxidative stress and inflammation induced by the inhalation of dust particles can cause vascular endothelial

damage, increased platelet activity, and enhanced coagulation, thereby promoting thrombus formation^{22, 23}. There is also evidence that high concentrations of dust particles can cause an increase in heart rate and blood pressure^{24, 25}, suggesting the plausibility of the associations of SDS events with stroke mortality and hypertensive heart disease mortality found in our study.

Existing epidemiological studies have not examined the effects of SDS events on mortality due to genitourinary, nervous, and digestive system diseases. Our study first found positive associations between SDS events and these mortality outcomes, though not statistically significant. Several epidemiological and experimental studies have investigated the impacts of SDS on genitourinary, nervous, and digestive health. For example, Herrera-Molina et al. found that exposure to SDS events was associated with an increased risk of hospitalizations from genitourinary diseases²⁶; a cohort study from Barcelona, Spain, reported a significant increase in gestational age following the Saharan dust episodes²⁷. By collecting blood and urine samples from people affected by the dust storm, Badeenezhad et al. measured biomarkers related to the central nervous system and found that PM₁₀ during SDS could cause neuron and astrocyte damage, leading to neuropsychiatric disorders²⁸. Cao et al. found that repeated exposure to fine dust particles could cause pathological changes in the stomach of rats²⁰. Our study results and previous findings provide evidence suggesting the potentially harmful impact of SDS events on mortality due to genitourinary, nervous, and digestive system diseases.

The differences of health effects between SDS events and conventional PM pollution events (that is, PM_{2.5} pollution events which happen on non-SDS event days) were confirmed by our results. For ischemic stroke, intracerebral hemorrhagic stroke, hypertensive heart disease, chronic lower respiratory disease and COPD, SDS events could trigger more serious impact than conventional PM pollution events. This may be due to the differences in constituents of PM between SDS events and conventional PM pollution events. Sources of PM during SDS include both anthropogenic and natural

sources, which is more complicated¹⁵. He et al.²⁹ reported that $PM_{2.5}$ during SDS had greater exacerbating effect on the lung eosinophilia of mice than $PM_{2.5}$ in hazy weather. This could result from the increase bioreactivity of $PM_{2.5}$ during SDS. Ho et al.³⁰ found that significant amounts of suspended dust particles during SDS provided platforms to intermix with chemicals on their surfaces. And these reactions may be an unrecognized source of toxic compositions, enhancing the $PM_{2.5}$ toxicity during SDS. Furthermore, fine dust particles can be elevated into the troposphere and travel thousands of kilometers,³¹ absorbing airborne pollutants from anthropogenic sources in industrial areas, microorganisms, and potential allergens, such as pollens, and increasing the adverse health effects of $PM_{2.5}$ during SDS^{5, 32}.

When accounting for $PM_{2.5}$ and $PM_{2.5-10}$, we still observed added effects of SDS events on mortality from ischemic stroke, chronic lower respiratory disease, and COPD. Similarly, Sun et al.³³ observed that heavy $PM_{2.5}$ pollution events, defined as daily average $PM_{2.5}$ concentration $\geq 75 \mu\text{g}/\text{m}^3$ for at least 3 days, had added effects on the circulatory (0.96%; 95% CI: 0.37%, 1.55%) and respiratory (0.55%; 95% CI: -0.52%, 1.63%) mortality. SDS events have coincided with high levels of $PM_{2.5-10}$ and $PM_{2.5}$ ³⁴. Experimental studies found that exposure to sustained high $PM_{2.5}$ concentrations could cause severe damage to multiple organs in mice, including cardiac fibrosis and myocardial hypertrophy^{35, 36}. In addition, SDS may evoke the worry and stress which have been shown to contribute to health symptoms of all body systems, including vasoconstriction and increased blood pressure at a cardiovascular level^{37, 38, 39, 40}.

This study has some limitations. First, exposure misclassification is possible as there was no universal definition for SDS events⁴¹. However, we considered the official sand-dust weather records, the PM_{10} concentration, and $PM_{2.5}/PM_{10}$ concentration ratio in identifying SDS events in this study; we also used several alternative definitions for SDS in sensitivity analysis, with the results pretty robust to the primary results. Second, the limited accessibility of mortality data hampered us to include more comprehensive and even national regions in China. However, to the best of our knowledge, this study

is the largest epidemiological study to investigate the mortality risks associated with SDS. Third, we found significant added effects of SDS on cardiorespiratory mortality when accounting for PM exposures. Still, this study cannot distinguish natural and anthropogenic sources of PM, which is also not the scope of this study. More studies are needed to explore SDS's independent health effects with natural and anthropogenic PM accounted for.

Findings from this study can provide implications for policymakers and the public. First, given SDS's broad and severe health impact, it is necessary to establish air quality guidelines and standards for SDS. Second, the health departments should allocate medical resources, especially for people in need, before SDS events. Finally, the public should be educated and informed about the potential health risk of SDS and adequate protective measures.

To summarize, this nationwide multicenter study showed that short-term exposure to SDS events is linked to increased mortality from many causes, particularly respiratory diseases. Public health policy against SDS should be implemented, as SDS presents adverse health risk in addition to conventional PM pollution.” (Lines 111-250)

References for responses:

1. Fussell J. C., Kelly F. J. Mechanisms underlying the health effects of desert sand dust. *Environ. Int.* **157**, 106790 (2021).
2. Aghababaeian H. *et al.* Global health impacts of dust storms: A systematic review. *Environ. Health Insights* **15**, 11786302211018390 (2021).
3. Sajani S. Z. *et al.* Saharan dust and daily mortality in Emilia-Romagna (Italy). *Occup. Environ. Med.* **68**, 446-451 (2011).
4. Lee H., Honda Y., Lim Y. H., Guo Y. L., Hashizume M., Kim H. Effect of Asian dust storms on mortality in three Asian cities. *Atmos. Environ.* **89**, 309-317 (2014).
5. Hashizume M. *et al.* Health effects of Asian dust: A systematic review and meta-analysis. *Environ. Health. Perspect.* **128**, 66001 (2020).

6. Tam W. W. S., Wong T. W., Wong A. H., Hui D. S. C. Effect of dust storm events on daily emergency admissions for respiratory diseases. *Respirology* **17**, 143-148 (2012).
7. Vodonos A. *et al.* The impact of desert dust exposures on hospitalizations due to exacerbation of chronic obstructive pulmonary disease. *Air Qual. Atmos. Health* **7**, 433-439 (2014).
8. Lorentzou C. *et al.* Extreme desert dust storms and COPD morbidity on the island of Crete. *Int. J. Chronic Obstruct. Pulm. Dis.* **14**, 1763 (2019).
9. Zhang X., Zhao L., Tong D., Wu G., Dan M., Teng B. A systematic review of global desert dust and associated human health effects. *Atmosphere* **7**, (2016).
10. Hsieh N. H., Liao C. M. Assessing exposure risk for dust storm events-associated lung function decrement in asthmatics and implications for control. *Atmos. Environ.* **68**, 256-264 (2013).
11. Eisenbarth S. C., Colegio O. R., O'Connor W., Sutterwala F. S., Flavell R. A. Crucial role for the Nalp3 inflammasome in the immunostimulatory properties of aluminium adjuvants. *Nature* **453**, 1122-1126 (2008).
12. Ichinose T. *et al.* Effects of asian sand dust, Arizona sand dust, amorphous silica and aluminum oxide on allergic inflammation in the murine lung. *Inhal. Toxicol.* **20**, 685-694 (2008).
13. Dorman D. C. *et al.* Biological responses in rats exposed to cigarette smoke and Middle East sand (dust). *Inhal. Toxicol.* **24**, 109-124 (2012).
14. Ghio A. J. *et al.* Biological effects of desert dust in respiratory epithelial cells and a murine model. *Inhal. Toxicol.* **26**, 299-309 (2014).
15. Li G. *et al.* Resurgence of sandstorms complicates China's air pollution situation. *Environ. Sci. Technol.* **55**, 11467-11469 (2021).
16. Ichinose T. *et al.* The effects of microbial materials adhered to Asian sand dust on allergic lung inflammation. *Arch. Environ. Contam. Toxicol.* **55**, 348-357 (2008).
17. Seaton A., MacNee W., Donaldson K., Godden D. Particulate air pollution and acute health effects. *Lancet* **345**, 176-178 (1995).

18. Matsukawa R. *et al.* Desert dust is a risk factor for the incidence of acute myocardial infarction in Western Japan. *Circ. Cardiovasc. Qual. Outcomes* **7**, 743-748 (2014).
19. Kamouchi M., Ueda K., Ago T., Nitta H., Kitazono T., Fukuoka Stroke Registry I. Relationship between asian dust and ischemic stroke: A time-stratified case-crossover study. *Stroke* **43**, 3085-3087 (2012).
20. Cao X. J. *et al.* Effects of dust storm fine particle-inhalation on the respiratory, cardiovascular, endocrine, hematological, and digestive systems of rats. *Chin. Med. J. (Engl.)* **131**, 2482-2485 (2018).
21. Yang C. Y., Chen Y. S., Chiu H. F., Goggins W. B. Effects of Asian dust storm events on daily stroke admissions in Taipei, Taiwan. *Environ. Res.* **99**, 79-84 (2005).
22. Brook R. D. *et al.* Particulate matter air pollution and cardiovascular disease: An update to the scientific statement from the American Heart Association. *Circulation* **121**, 2331-2378 (2010).
23. Piepoli M. F. *et al.* 2016 European Guidelines on cardiovascular disease prevention in clinical practice: The Sixth Joint Task Force of the European Society of Cardiology and Other Societies on Cardiovascular Disease Prevention in Clinical Practice (constituted by representatives of 10 societies and by invited experts)Developed with the special contribution of the European Association for Cardiovascular Prevention & Rehabilitation (EACPR). *Eur. Heart J.* **37**, 2315-2381 (2016).
24. Lipsett M. J., Tsai F. C., Roger L., Woo M., Ostro B. D. Coarse particles and heart rate variability among older adults with coronary artery disease in the Coachella Valley, California. *Environ. Health Perspect.* **114**, 1215-1220 (2006).
25. Chang C. C., Hwang J. S., Chan C. C., Wang P. Y., Cheng T. J. Effects of concentrated ambient particles on heart rate, blood pressure, and cardiac contractility in spontaneously hypertensive rats during a dust storm event. *Inhal. Toxicol.* **19**, 973-978 (2007).
26. Herrera-Molina E., Gill T. E., Ibarra-Mejia G., Jeon S. Associations between dust exposure and hospitalizations in El Paso, Texas, USA. *Atmosphere* **12**, (2021).

27. Dadvand P. *et al.* Saharan dust episodes and pregnancy. *J. environ. Monit.* **13**, 3222-3228 (2011).
28. Badeenezhad A. *et al.* Investigating the relationship between central nervous system biomarkers and short-term exposure to PM₁₀-bound metals during dust storms. *Atmos. Pollut. Res.* **11**, 2022-2029 (2020).
29. He M. *et al.* Differences in allergic inflammatory responses between urban PM_{2.5} and fine particle derived from desert-dust in murine lungs. *Toxicol. Appl. Pharm.* **297**, 41-55 (2016).
30. Ho K. F. *et al.* Contributions of local pollution emissions to particle bioreactivity in downwind cities in China during Asian dust periods. *Environ. Pollut.* **245**, 675-683 (2019).
31. Middleton N., Kang U. Sand and dust storms: Impact mitigation. *Sustainability* **9**, 1053 (2017).
32. Wu Y., Wen B., Li S., Guo Y. Sand and dust storms in Asia: A call for global cooperation on climate change. *Lancet Planet. Health* **5**, e329-e330 (2021).
33. Sun Y. *et al.* Impact of heavy PM_{2.5} pollution events on mortality in 250 Chinese counties. *Environ. Sci Technol.* **56**, 8299-8307 (2022).
34. WHO. WHO global air quality guidelines: particulate matter (PM_{2.5} and PM₁₀), ozone, nitrogen dioxide, sulfur dioxide and carbon monoxide (2021).
35. Li D. *et al.* Multiple organ injury in male C57BL/6J mice exposed to ambient particulate matter in a real-ambient PM exposure system in Shijiazhuang, China. *Environ. Pollut.* **248**, 874-887 (2019).
36. Su X. *et al.* Ambient PM_{2.5} caused cardiac dysfunction through FoxO1-targeted cardiac hypertrophy and macrophage-activated fibrosis in mice. *Chemosphere* **247**, 125881 (2020).
37. Stenlund T., Lidén E., Andersson K., Garvill J., Nordin S. Annoyance and health symptoms and their influencing factors: A population-based air pollution intervention study. *Public Health* **123**, 339-345 (2009).

38. Claeson A. S., Lidén E., Nordin M., Nordin S. The role of perceived pollution and health risk perception in annoyance and health symptoms: A population-based study of odorous air pollution. *Int. Arch. Occup. Environ. health* **86**, 367-374 (2013).
39. Orru K., Nordin S., Harzia H., Orru H. The role of perceived air pollution and health risk perception in health symptoms and disease: A population-based study combined with modelled levels of PM₁₀. *Int. Arch. Occup. Environ. Health* **91**, 581-589 (2018).
40. Kwon H. J., Cho S. H., Chun Y., Lagarde F., Pershagen G. Effects of the Asian dust events on daily mortality in Seoul, Korea. *Environ. Res.* **90**, 1-5 (2002).
41. Hoffmann C., Funk R., Wieland R., Li Y., Sommer M. Effects of grazing and topography on dust flux and deposition in the Xilingele grassland, Inner Mongolia. *J. Arid Environ.* **72**, 792-807 (2008).

Minor comments:

- Authors should limit the use of acronyms. At present, they have too many and it is challenging for the reader to follow.

Response:

Thanks for identifying this issue. We modified the acronyms and retained some widely known terms, such as COPD, in the revised manuscript.

Responses to Reviewer #2:

This manuscript evaluated the cause-specific mortality risks associated with sand and storms in China. The strength is the use of a multicenter dataset comprising of 214 counties. This topic is not novel, given that many previous studies worldwide had examined these associations, albeit the number of death causes previously evaluated are limited.

Response:

We appreciate your time and efforts in reviewing our manuscript and your constructive suggestions. We also strongly agree with you that considerable studies have been conducted to examine the health effects of SDS events. However, the latest meta-

analysis¹ and review²⁻⁵ that focused on the health effects of SDS published in *Environmental Health Perspectives* etc. all show that the available studies only can provide evidence that SDS events are associated with mortality risks from all-cause, the overall circulatory and respiratory diseases. For other diseases, including cardiopulmonary sub-causes, such as ischemic heart disease, acute myocardial infarction, and chronic obstructive pulmonary disease, existing studies have focused on the investigation of the effect of SDS on morbidity, and mortality-related evidence is still lacking. Also, most relevant studies are conducted at a single location with relatively small sample sizes. The lack of a national assessment of SDS' health impact impedes the science-based national and regional cooperation to mitigate and cope with the adverse effects of SDS.

In 2021, World Health Organization (WHO) updated the global air quality guidelines (AQG) and noted that it has long wanted to address SDS in this update and formulate an AQG level for SDS since 2016 due to health concerns. However, this action was hindered by insufficient epidemiological evidence on independent adverse health effects from SDS. There are still no studies that have systematically investigated the risk of death from SDS events since the above meta-analysis and review articles were published. This has led to an incomplete chain of evidence for the health effects of SDS up to now.

In the context of climate change, the problems associated with SDS are further accentuated due to intensified droughts and greater occurrence of wind erosion and extreme weather events, such as heatwave and drought^{6,7}. For example, China, a major dust source region in the world, experienced one of the strongest dust weather processes in the past decade on March 15, 2021, affecting 12 provinces and nearly one-third of the country's land^{6,8}. At the same time, there are fewer studies related to the health effects of SDS in China, which leads to a particular lack of clarity on the health effects of SDS in China and makes it difficult to support relevant public health protection actions. In this context, there is an urgent need to conduct a systematic investigation of

the mortality risk of SDS in China based on national, large-sample population data, with China as the study area.

References for responses:

1. Hashizume, M. *et al.* Health effects of Asian dust: A systematic review and meta-analysis. *Environ. Health Persp.* **128**, 066001 (2020).
2. Hwong, A. R. *et al.* Climate change and mental health research methods, gaps, and priorities: A scoping review. *Lancet Planet. Health* **6**, e281–e291 (2022).
3. Aghababaeian, H. *et al.* Global health impacts of dust storms: A systematic review. *Environ. Health Insights* **15**, 11786302211018390 (2021).
4. Sadeghimoghaddam, A., Khankeh, H., Norozi, M., Fateh, S. & Farrokhi, M. Investigating the effects of dust storms on morbidity and mortality due to cardiovascular and respiratory diseases: A systematic review. *J. Educ. Health Promot.* **10**, 191 (2021).
5. Zhang, X. *et al.* A systematic review of global desert dust and associated human health effects. *Atmosphere* **7**, 158 (2016).
6. Wu, Y., Wen, B., Li, S. & Guo, Y. Sand and dust storms in Asia: A call for global cooperation on climate change. *Lancet Planet. Health* **5**, e329–e330 (2021).
7. Bayram, H. *et al.* Environment, Global climate change, and cardiopulmonary health. *Am. J. Respir. Crit. Care Med.* **195**, 718–724 (2017).
8. Filonchyk, M. Characteristics of the severe March 2021 Gobi Desert dust storm and its impact on air pollution in China. *Chemosphere* **287**, 132219 (2022).

Some specific concerns are listed below.

1. Line 69, why the number of counties used for the pool estimates of each death cause was >100? Please give more details.

Response:

Thanks for your professional comment. We apologize for the unclear description in the original manuscript. We intended to describe the number of counties included in calculating the pooled estimates for each mortality outcome to reflect the

representativeness of the pooled estimates. To avoid further misunderstanding caused to readers, we have removed relevant sentences in the revised manuscript.

2. Abbreviations in the results should be defined when they are firstly presented.

Response:

Thanks for identifying this issue. We have thoroughly modified abbreviations in our manuscript and provided the full term at the first mention.

3. Line 99, a total of 1,495,724 deaths were used. It is unclear on whether they occurred during the SDS event periods? If they covered the whole year, this number is useless as only the SDS event period was explored.

Response:

Thanks for pointing out this issue. The originally reported number (i.e. 1,495,724) was the deaths occurred during the SDS period (1 February-31 May) during 2013 and 2018. Also, as a response to other comment from reviewer 1, we also added death numbers occurred during the SDS event days and edited relevant text in the revised manuscript.

Relevant text in revised Results section:

“A total of 1,495,724 deaths occurred during the SDS period, and 19,082 deaths in the identified SDS event days (Table 2).” (Lines 60-61)

Table 2. Summary statistics for daily mortality in study counties during the SDS period (1 February-31 May) from 2013 to 2018

ICD-10	Cause of mortality	Total deaths during the SDS period	Mean daily deaths during the SDS period (range)	Total deaths occurred on SDS event days
A00-Z99	All (ALL)	1,495,724	12.5 (0,116)	19,082
A00-R99	Non-accidental (TOTAL)	1,406,898	11.7 (0,116)	17,923

Broad category

I00-I99	Diseases of the circulatory system (CIR)	666,474	5.6 (0,110)	9,066
J00-J99	Diseases of the respiratory system (RES)	173,632	1.6 (0,24)	2,156
K00-K93	Diseases of the digestive system (DSD)	26,504	0.3 (0,6)	321
G00-G99	Diseases of the nervous system (NSD)	21,816	0.2 (0,9)	185
N00-N99	Diseases of the genitourinary system (GSD)	13,452	0.2 (0,6)	181

Specific category

I00-I99 Diseases of the circulatory system (CIR)

I20-I25	Ischemic heart disease (IHD)	268,305	2.5 (0,43)	3,896
I20-I22, I24	Acute ischemic heart disease (AIHD)	152,585	1.5 (0,27)	2,518
I21-I23	Myocardial infarction (MI)	151,335	1.5 (0,27)	2,459
I21-I22	Acute Myocardial infarction (AMI)	151,264	1.5 (0,27)	2,456
I25	Chronic ischemic heart disease (CIHD)	125,496	1.3 (0,36)	1,554
I63	Ischemic stroke (ISTR)	114,376	1.1 (0,36)	1,487
I60-I61	Hemorrhagic stroke (HSTR)	101,592	1.0 (0,34)	1,486
I61	Intracerebral hemorrhagic stroke (IHDSTR)	94,827	0.9 (0,34)	1,400
I10-I15	Hypertensive heart disease (HBP)	41,640	0.5 (0,19)	636

J00-J99 Diseases of the respiratory system (RES)

J40-J47	Chronic lower respiratory disease (CLRI)	124,744	1.2 (0,23)	1,571
J41-J44	Chronic obstructive pulmonary disease (COPD)	119,944	1.1 (0,23)	1,514

Relevant text in revised Discussion section:

“In this nationwide study, we comprehensively investigated the cause-specific mortality risks of short-term exposure to SDS events using the data of 1,495,724 deaths from 214 Chinese counties during the SDS period between 2013 and 2018.” (Lines 111-113)

4. Line 250, “all analyses were restricted to the period with a high frequency of SDS events during the study”, why only February to May was included? From Figure S3, some other seasons also had SES events.

Response:

Thanks for your professional suggestion. In this study, we defined the period from February to May, when SDS frequently occurs, as the SDS period. We examined the

health effects of SDS events using daily data for the SDS period in the primary analysis. This approach is widely used in epidemiologic studies about climate extreme events. Climate extremes events, such as heat waves, cold spells, and tropical cyclones, typically occur within specific seasons. To investigate the short-term health effects of climate extreme events, most researchers used time-series analysis using data restricted for specific periods¹⁻⁸, as used in the present study; for example, examining health effects of heat waves has been analyzed using data for the warm season. Some other researchers employed a case-crossover design or matched analysis (similar to case-crossover)^{9,10}. Analyzing data restricted for specific periods rather than the whole year can eliminate any confounding bias that could arise from factors varying across the whole year in a computationally efficient way.

Also, to investigate if the effect estimates would differ when fitting the model using data for the SDS period or the whole year, we conducted a sensitivity analysis by running the same regression model with data for the whole year. Results were pretty consistent with those from the primary analysis, that is, fitting the model with data for the SDS period. We also revised the corresponding text in the manuscript.

Relevant text in revised Methods section:

“Forth, we refit the GLM model in the first stage using the whole year’s data.” (Lines 355-356)

Relevant text in revised Results section:

“Results from sensitivity analyses, by changing the degree of freedom of spline functions, changing the adjustment of meteorological parameters (Fig. S6), and using the data of different study periods (Fig. S7), generally remained consistent with those from the main models for most mortality outcomes.” (Lines 104-108)

Fig. S7. Excess risk (ER, %) and 95% confidence interval for mortality associated with sand and dust storms (SDS) events based on models fit with different study period. “Study period 1” represents estimates from our main analysis conducted during the SDS period (1 February–31 May), 2013-2018. “Study period 2” represents estimates from the main model equation conducted during the whole year, 2013-2018.

References for responses:

1. Wang, Q. *et al.* Independent and combined effects of heatwaves and PM_{2.5} on preterm birth in guangzhou, China: A survival analysis. *Environ. Health Perspectives* **128**, 017006 (2020).
2. Sun, Z. *et al.* Heat wave characteristics, mortality and effect modification by temperature zones: A time-series study in 130 counties of China. *Int. J. Epidemiol.* **49**, 1813–1822 (2021).
3. Zhao, Q. *et al.* The association between heatwaves and risk of hospitalization in Brazil: A nationwide time series study between 2000 and 2015. *PLoS Med.* **16**,

- e1002753 (2019).
4. Guo, Y. *et al.* Quantifying excess deaths related to heatwaves under climate change scenarios: A multicountry time series modelling study. *PLoS Med.* **15**, e1002629 (2018).
 5. Anderson, G. B. & Bell, M. L. Heat waves in the United States: Mortality risk during heat waves and effect modification by heat wave characteristics in 43 U.S. communities. *Environ. Health Persp.* **119**, 210–218 (2011).
 6. Hansen, A. *et al.* The effect of heat waves on mental health in a temperate Australian city. *Environ. Health Persp.* **116**, 1369–1375 (2008).
 7. Ma, C. *et al.* Cold spells and cause-specific mortality in 47 Japanese prefectures: A systematic evaluation. *Environ. Health Persp.* **129**, 067001.
 8. Hansen, A. L. *et al.* The effect of heat waves on hospital admissions for renal disease in a temperate city of Australia. *Int. J. Epidemiol.* **37**, 1359–1365 (2008).
 9. Parks, R. M. *et al.* Tropical cyclone exposure is associated with increased hospitalization rates in older adults. *Nat. Commun.* **12**, 1545 (2021).
 10. Yan, M. *et al.* Cardiovascular mortality risks during the 2017 exceptional heatwaves in China. *Environ. Int.* **172**, 107767 (2023).

5. More importantly, the first-stage analysis was conducted at the county level. However, as showed in Table 1, the daily cases for many specific causes of deaths were quite few (equal or lower than 1 case), which would greatly attenuate the statistical power in an ecological time-series analysis.

Response:

Thanks for your comment. We agree that the small number of some specific causes of death at the county level may affect statistical power. Yet, time series studies at the county level have the advantage of enabling exposure measurement as accurately as possible based on the most minor administrative units and is a typical environmental epidemiological study design^{1–9}; for example, the well-known NMMAPS (National Morbidity and Mortality Air Pollution Study) dataset collects mortality data at the county level in the U.S. and has been applied in various time series studies, yielding

abundant and substantial adverse health impact evidence of air pollution¹⁰⁻¹⁹. The daily mean death count at the county level is 8.2 in the NMMAPS study, which is lower than the daily mean death count at the county level in our study (11.7). Thus, a time-series ecological study using mortality data at the county level is feasible.

Moreover, our county-level mortality data merits a long time series (6 years) and enough counties (214 counties). And we applied a two-stage time-series analysis approach. Our mortality data is the largest sample of good quality currently available. At the same time, our team has conducted multiple studies on specific causes of death at the county level in China, including studies using the same county-level mortality data used in this study, published in *Science Advances*, *Environmental Health Perspectives*, etc., knowing that the number of cases of death >100 per year for four years can provide sufficient statistical validity²⁰⁻³². The results of our study also showed good statistical validity, finding a significant effect of SDS events on diseases of a specific category, including ischemic stroke, intracerebral hemorrhagic stroke, hypertensive heart disease, myocardial infarction, acute myocardial infarction, and acute ischemic heart disease, chronic lower respiratory disease and chronic obstructive pulmonary disease. Thus, we believe that the present results are plausible. To communicate clearly, we have added relevant text in the revised manuscript.

Relevant text in revised Methods section:

“In the first stage, we fit a generalized linear model (GLM) with quasi-Poisson distribution to assess the effects of SDS on mortality from a spectrum of causes for each county. This county level analysis based our mortality data have been shown feasible enough to ensure statistical power by previous researches¹⁻¹⁹ and our team's researches²⁰⁻³².” (Lines 308-312)

References for responses:

1. Ban, J., Su, W., Zhong, Y., Liu, C. & Li, T. Ambient formaldehyde and mortality: A time series analysis in China. *Sci. Adv.* **8**, eabm4097 (2022).

2. Anderson, G. B. *et al.* Assessing United States county-level exposure for research on tropical cyclones and human health. *Environ. health persp.* **128**, 107009. (2020).
3. Khatana, S. A. M., Werner, R. M., & Groeneveld, P. W. Association of extreme heat and cardiovascular mortality in the United States: A county-level longitudinal analysis from 2008 to 2017. *Circulation* **146**, 249-261 (2022).
4. Parks, R. M. *et al.* Association of tropical cyclones with county-level mortality in the US. *J. Am. Med. Assoc.* **327**, 946–955 (2022).
5. Sun, Z. *et al.* Heat wave characteristics, mortality and effect modification by temperature zones: A time-series study in 130 counties of China. *Int. J. Epidemiol.* **49**, 1813-1822 (2020).
6. Pye, H. O. T. *et al.* Secondary organic aerosol association with cardiorespiratory disease mortality in the United States. *Nat. Commun.* **12**, 7215 (2021).
7. He, M. Z. *et al.* Short-and intermediate-term exposure to NO₂ and mortality: A multi-county analysis in China. *Environ. Pollut.* **261**, 114165 (2020).
8. Crooks, J. L. *et al.* The association between dust storms and daily non-accidental mortality in the United States, 1993–2005. *Environ. Health Persp.* **124**, 1735-1743 (2016).
9. Chen, K. *et al.* Urbanization level and vulnerability to heat-related mortality in Jiangsu Province, China. *Environ. Health Persp.* **124**, 1863–1869 (2016).
10. Welty, L. J., Peng, R. D., Zeger, S. L. & Dominici, F. Bayesian distributed lag models: Estimating effects of particulate matter air pollution on daily mortality. *Biometrics* **65**, 282–291 (2009).
11. Lim, Y. H., Reid, C. E., Mann, J. K., Jerrett, M. & Kim, H. Diurnal temperature range and short-term mortality in large US communities. *Int. J. Biometeorol.* **59**, 1311–1319 (2015).
12. Roberts, S. & Martin, M. A. Methods for bias reduction in time-series studies of particulate matter air pollution and mortality. *J. Toxicol. Env. Health Part A* **70**, 665–675 (2007).
13. Wilson, A., Rappold, A. G., Neas, L. M. & Reich, B. J. Modeling the effect of temperature on ozone-related mortality. *Ann. Appl. Stat.* **8**, 1728–1749 (2014).

14. Zhang, Y. *et al.* Mortality risk and burden associated with temperature variability in China, United Kingdom and United States: Comparative analysis of daily and hourly exposure metrics. *Environ. Res.* **179**, 108771 (2019).
15. Ren, C., Williams, G. M., Morawska, L., Mengersen, K. & Tong, S. Ozone modifies associations between temperature and cardiovascular mortality: Analysis of the NMMAPS data. *Occup. Environ. Med.* **65**, 255–260 (2008).
16. Smith, R. L., Xu, B. & Switzer, P. Reassessing the relationship between ozone and short-term mortality in US urban communities. *Inhal. Toxicol.* **21**, 37–61 (2009).
17. Zhang, Y. *et al.* Socio-geographic disparity in cardiorespiratory mortality burden attributable to ambient temperature in the United States. *Environ. Sci. Pollut. Res.* **26**, 694–705 (2019).
18. Ren, C., Williams, G. M., Mengersen, K., Morawska, L. & Tong, S. Temperature enhanced effects of ozone on cardiovascular mortality in 95 large US communities, 1987–2000: Assessment using the NMMAPS Data. *Arch. Environ. Occup. Health* **64**, 177–184 (2009).
19. Guo, B. *et al.* Using spatio-temporal lagged association pattern to unravel the acute effect of air pollution on mortality. *Sci. Total Environ.* **664**, 99–106 (2019).
20. Ban, J., Su, W., Zhong, Y., Liu, C. & Li, T. Ambient formaldehyde and mortality: A time series analysis in China. *Sci. Adv.* **8**, eabm4097 (2022).
21. Yan, M. *et al.* Cardiovascular mortality risks during the 2017 exceptional heatwaves in China. *Environ. Int.* **172**, 107767 (2023).
22. Zhong, Y., Chen, C., Wang, Q. & Li, T. High temperature and risk of cause-specific mortality in China, 2013–2018. *CCDCW* **2**, 408–412 (2020).
23. Liu, Y. *et al.* Increased mortality risks from a spectrum of causes of tropical cyclone exposure — China, 2013–2018. *CCDCW* **5**, 119–124 (2023).
24. Li, T., Yan, M., Sun, Q. & Anderson, G. B. Mortality risks from a spectrum of causes associated with wide-ranging exposure to fine particulate matter: A case-crossover study in Beijing, China. *Environ. Int.* **111**, 52–59 (2018).
25. Chen, C. *et al.* Short-term exposure to fine particles and risk of cause-specific mortality — China, 2013–2018. *CCDCW* **1**, 8–12 (2019).

26. Chen, C. *et al.* Short-term exposure to ozone and cause-specific mortality risks and thresholds in China: Evidence from nationally representative data, 2013-2018. *Environ. Int.* **171**, 107666 (2023).
27. Chen, C. *et al.* Short-term exposures to PM_{2.5} and cause-specific mortality of cardiovascular health in China. *Environ. Res.* **161**, 188–194 (2018).
28. Du, P. *et al.* Traffic-related PM_{2.5} and its specific constituents on circulatory mortality: A nationwide modelling study in China. *Environ. Int.* **170**, 107652 (2022).
29. Sun, Q. *et al.* Health risks and economic losses from cold spells in China. *Sci. Total Environ.* **821**, 153478 (2022).
30. Sun, Y. *et al.* Impact of heavy PM_{2.5} pollution events on mortality in 250 Chinese counties. *Environ. Sci. Technol.* **56**, 8299–8307 (2022).
31. Shi, W. *et al.* Modification effects of temperature on the ozone–mortality relationship: a nationwide multicounty study in China. *Environ. Sci. Technol.* **54**, 2859-2868 (2020).
32. He, M. Z. *et al.* Short-and intermediate-term exposure to NO₂ and mortality: A multi-county analysis in China. *Environ. Pollut.* **261**, 114165 (2020).

6. Line 262, only 2 df per year for time trends may be not sufficient.

Response:

Thanks for your valuable suggestion. The 2 df was applied per SDS period (1 February - 31 May) for time trends. We analyzed the health effects of SDS events using data for the SDS period rather than the whole year. This approach has also been used in previous studies on the health effects of other climate extreme events, such as heat waves and cold spells¹⁻⁸, often limiting the analysis to the warm or cold season. We also used a greater df (3 df) per SDS period for time trends in the sensitivity analysis to investigate the robustness of our results. We also modified relevant text and provided a more precise description of the selection of df.

Relevant text in revised Methods section:

“We used the natural spline functions (“ns”) in controlling the confounding of long time trend, daily mean temperature (“Temp_t”), and daily relative humidity (“RH_t”), with the degrees of freedom (df) of 2 (per SDS period), 3, and 3, respectively.” (Lines 321-324)

References for responses:

1. Wang, Q. *et al.* Independent and combined effects of heatwaves and PM_{2.5} on preterm birth in Guangzhou, China: A survival analysis. *Environ. Health Persp.* **128**, 017006 (2020).
2. Sun, Z. *et al.* Heat wave characteristics, mortality and effect modification by temperature zones: A time-series study in 130 counties of China. *Int. J. Epidemiol.* **49**, 1813–1822 (2021).
3. Zhao, Q. *et al.* The association between heatwaves and risk of hospitalization in Brazil: A nationwide time series study between 2000 and 2015. *PLoS Med.* **16**, e1002753 (2019).
4. Guo, Y. *et al.* Quantifying excess deaths related to heatwaves under climate change scenarios: A multicountry time series modelling study. *PLoS Med.* **15**, e1002629 (2018).
5. Anderson, G. B. & Bell, M. L. Heat waves in the United States: Mortality risk during heat waves and effect modification by heat wave characteristics in 43 U.S. communities. *Environ. Health Persp.* **119**, 210–218 (2011).
6. Hansen, A. *et al.* The effect of heat waves on mental health in a temperate Australian city. *Environ. Health Persp.* **116**, 1369–1375 (2008).
7. Ma, C. *et al.* Cold spells and cause-specific mortality in 47 Japanese prefectures: A systematic evaluation. *Environ. Health Persp.* **129**, 067001.
8. Hansen, A. L. *et al.* The effect of heat waves on hospital admissions for renal disease in a temperate city of Australia. *Int. Journal Epidemiol.* **37**, 1359–1365 (2008).
7. Line 264, the lagged effects of weather conditions should be also controlled. 3 df for temperature is also not enough.

Response:

Thanks for your valuable suggestion. In the revised manuscript, we have controlled the lagged effects of weather conditions. Specifically, we adjusted for the lagged effects of temperature and relative humidity in the regression models when estimating the lagged mortality risks of SDS events. Moreover, we conducted a sensitivity analysis with adjustments of the 21-day moving average of temperature and the 7-day moving average of relative humidity¹. The results corroborate the robustness of our results.

The df for temperature was selected based on existing epidemiologic studies on the health effects of climate extreme events, such as heat waves, cold spells²⁻⁹; these studies often limit the analysis to the specific periods that these climate extreme events typically occur (i.e., the warm or cold season), with 3 df typically used for weather variables. Since we also restricted the analysis to the SDS period (1 February - 31 May), 3 df for temperature was then applied in our study. We also used a greater degree of freedom (i.e., 4 df, 5 df) for weather variables in the sensitivity analyses, with results being consistent with our primary results.

Relevant text in revised Methods section:

“In addition, to investigate the potential delayed effects of SDS events exposure, we performed lagged analysis by fitting the same model separately for three single-day lagged data (lag 1, 2, and 3). For example, analysis at lag 1 estimates the impact on mortality on day t (Y_t) associated with exposure to the previous day (i.e., Z_{t-1}). In the lagged analysis, we used data for the same lagged day for $Temp_t$ and RH_t .” (Lines 341-345)

“Second, we changed the df for the time trend variable ($df = 3$) and used two different df ($df = 4, 5$) for meteorological parameters in the spline functions. Third, instead of the daily mean temperature and relative humidity, we used the 21-day moving average of temperature and the 7-day moving average of relative humidity to fully adjust for the confounding of meteorological conditions¹.” (Lines 351-355)

Relevant text in revised Results section:

“Results from sensitivity analyses, by changing the degree of freedom of spline functions, changing the adjustment of meteorological parameters (Fig. S6), and using the data of different study periods (Fig. S7), generally remained consistent with those from the main models for most mortality outcomes.” (Lines 104-108)

Fig. S6. Excess risk (ER, %) and 95% confidence interval for mortality associated with sand and dust storms events using different model settings. “Model 1” represents estimates from the primary model, with the degree of freedom (df) of 2 and 3, for time variable and meteorological parameters in natural spline functions. “Model 2” represents estimates from the primary model, except the df for the time variable was 3. “Model 3” represents estimates from the primary model, except the df for the time variable was 3, and the dfs for meteorological parameters were 4. “Model 4” represents estimates from the primary model, except the df for the time variable was 3, and the dfs for meteorological parameters were 5. “Model 5” represents estimates from the primary model, except using a 21-day moving average

of temperature and a 7-day moving average of relative humidity. Red represents that the ER differed significantly from 0% ($P < 0.05$). The Mortality of broad causes results are shown in the top two panels, and mortality of specific causes results in the bottom two panels.

References for responses:

1. Ye, T. *et al.* Risk and burden of hospital admissions associated with wildfire-related PM_{2.5} in Brazil, 2000–15: A nationwide time-series study. *Lancet Planet. Health*, **5**, e599-e607 (2021).
2. Wang, Q. *et al.* Independent and combined effects of heatwaves and PM_{2.5} on preterm birth in Guangzhou, China: A survival analysis. *Environ. Health Persp.* **128**, 017006 (2020).
3. Sun, Z. *et al.* Heat wave characteristics, mortality and effect modification by temperature zones: a time-series study in 130 counties of China. *Int. J. Epidemiol.* **49**, 1813–1822 (2021).
4. Zhao, Q. *et al.* The association between heatwaves and risk of hospitalization in Brazil: A nationwide time series study between 2000 and 2015. *PLoS Med.* **16**, e1002753 (2019).
5. Guo, Y. *et al.* Quantifying excess deaths related to heatwaves under climate change scenarios: A multicountry time series modelling study. *PLoS Med.* **15**, e1002629 (2018).
6. Anderson, G. B. & Bell, M. L. Heat waves in the United States: Mortality risk during heat waves and effect modification by heat wave characteristics in 43 U.S. communities. *Environ. Health Persp.* **119**, 210–218 (2011).
7. Hansen, A. *et al.* The effect of heat waves on mental health in a temperate Australian city. *Environ. Health Persp.* **116**, 1369–1375 (2008).
8. Ma, C. *et al.* Cold spells and cause-specific mortality in 47 Japanese prefectures: A systematic evaluation. *Environ. Health Persp.* **129**, 067001.
9. Hansen, A. L. *et al.* The effect of heat waves on hospital admissions for renal disease in a temperate city of Australia. *Int. J. Epidemiol.* **37**, 1359–1365 (2008).

8. Line 278, “Series of sensitivity analyses” should be also specified.

Response:

We have further described the sensitivity analyses in detail.

Relevant text in revised Methods section:

“Sensitivity analyses

We conducted a series of sensitivity analyses to assess the robustness of the estimated associations between SDS events and many mortality outcomes in our primary analysis. First, we examined the mortality risk of SDS events under different SDS definitions. Second, we changed the df for the time trend variable (df = 3) and used two different df (df = 4, 5) for meteorological parameters in the spline functions. Third, instead of the daily mean temperature and relative humidity, we used the 21-day moving average of temperature and the 7-day moving average of relative humidity to fully adjust for the confounding of meteorological conditions¹. Forth, we refit the GLM model in the first stage using the whole year’s data.” (Lines 347-356)

References for responses:

1. Ye, T. *et al.* Risk and burden of hospital admissions associated with wildfire-related PM_{2.5} in Brazil, 2000–15: A nationwide time-series study. *Lancet Planet. Health*, **5**, e599-e607 (2021).

Reviewer #3 (Remarks to the Author):

In this paper the authors aim to produce novel evidence on the association between cause-specific mortality and sand and dust storms in China. The paper has the merit to collect a substantial amount of data from multiple areas of China. However, there are several flaws in the statement of the objectives, the details of the statistical analysis, the wording of many sentences (sometimes truncated), and the interpretation of the results.

Response:

Thanks for your professional and constructive suggestions. First, we agree our original

manuscript did not clearly describe the contribution of our findings to the existing literature. We modified the description of the objectives of this study in the last paragraph of the Introduction:

“Here, we conducted a nationwide multicenter time series study in China. Our objectives were to: (1) investigate the overall short-term effects of SDS events on mortality from a series of causes, identifying the spectrum of SDS-sensitive health outcome; (2) explore the added short-term effects of SDS events on mortality. Findings from this study will improve current understanding of the health effects of SDS.” (Lines 48-52)

Second, according to these issues raised from your comments, we realized the description of the statistical analysis might not be clear enough. For example, the degree of freedom values used in our analysis differed from previous time-series studies using whole year’s data. In the revised manuscript, we have comprehensively modified the description of the methods.

Finally, the interpretation of the results seemed problematic in the original manuscript. We agreed that the effect estimates of SDS events with and without adjustment for PM did not say anything about “the contribution of PM to SDS’s health effects”. Instead, effect estimates from the model with no PM adjusted represent the overall effects of SDS events, and effect estimates from the model with PM adjusted represent the added effects of SDS events. Thus, we removed this part and modified the relevant text in the revised manuscript.

We have thoroughly revised the manuscript and responded to your concerns raised from your comments, with a point-by-point response provided below. We hope you will be more enthusiastic about our manuscript after we have addressed these issues from your comments.

First, the paper mentions a spectrum of causes in the title, and indeed evaluates many of them, but little is reported in the interpretation of results in terms of cause-specific effect estimates. Adverse effects of dust on CVD and RESP mortality are largely known, and it is not clear how the paper contributes on the topic.

Response:

Thanks for your comment. In our revised manuscript, we provided interpretations of effect estimates for cause-specific mortality, such as ischemic stroke, hypertensive heart disease, and COPD mortality. Although several studies have provided evidence on the effect of SDS events on overall CVD and RESP mortality, there is a lack of evidence related to mortality due to cardiopulmonary sub-causes. This study, for the first time, reported positive associations of SDS events with mortality due to ischemic stroke, intracerebral hemorrhagic stroke, hypertensive heart disease, myocardial infarction, acute myocardial infarction, acute ischemic heart disease, chronic lower respiratory disease, and COPD.

Further, we found significant independent added effects of SDS events for ischemic stroke, chronic lower respiratory disease, and COPD mortality. Finally, with nationwide mortality data for many causes, we examined the effects of SDS events on mortality from genitourinary, nervous, and digestive system diseases, which have not been reported in existing epidemiologic studies. We thoroughly modified the relevant text in the revised manuscript.

Relevant text in revised Abstract section:

“Sand and Dust Storms (SDS) pose considerable health risks worldwide, but little is known about the impacts of SDS on cause-specific mortality. This nationwide multicenter time-series study aimed to examine SDS-associated mortality risks extensively. We analyzed 1,495,724 deaths and 2,024 SDS events from 1st February to 31st May (2013–2018) in 214 Chinese counties. The excess mortality risks associated with SDS were 7.49% (95% CI: 3.12–12.05%), 5.40% (1.25–9.73%), 4.05% (0.41–7.83%), 3.45% (0.34–6.66%), 3.37% (0.28–6.55%), 3.33% (0.07–6.70%), 8.90%

(4.96–12.98%), 12.51% (6.31–19.08%), and 11.55% (5.55–17.89%) for ischemic stroke, intracerebral hemorrhagic stroke, hypertensive heart disease, myocardial infarction, acute myocardial infarction, acute ischemic heart disease, respiratory disease, chronic lower respiratory disease, and chronic obstructive pulmonary disease (COPD), respectively. SDS had significantly added effects on ischemic stroke, chronic lower respiratory disease, and COPD mortality. Our results suggest the need to implement public health policy against SDS.” (Lines 5-18)

Relevant text in revised Discussion section:

“To the best of our knowledge, this study is the first to elucidate the mortality risks of SDS using a large sample size and a spectrum of mortality outcomes. Respiratory mortality significantly and substantially increased during SDS event days compared with clean days (8.90%; 95% CI: 4.96%, 12.98%). We identified a spectrum of SDS-sensitive health outcomes, including ischemic stroke mortality, intracerebral hemorrhagic stroke mortality, hypertensive heart disease mortality, myocardial infarction mortality, acute myocardial infarction mortality, acute ischemic heart disease mortality, chronic lower respiratory disease mortality, and COPD mortality. Added effects of SDS events were observed for mortality due to ischemic stroke, chronic lower respiratory disease, and COPD. Findings from this study provided scientific evidence to deepen the current understanding of SDS’s health effects and to plan interventions to protect the public against SDS.” (Lines 113-125)

“For cause-specific respiratory mortality outcomes, we found COPD mortality substantially increased (11.55%; 95% CI: 5.55%, 17.89%) during SDS, indicating COPD patients could be highly vulnerable to SDS’s adverse effects. This finding was consistent with previous relevant studies analyzing the effects of SDS on respiratory morbidity^{1, 2, 3}; for example, a study estimated COPD hospitalization increased by 16% (95% CI: 8%, 24%) during SDS in a city of Southern Israel². The elevated respiratory mortality risk associated with SDS may be due to considerable dust particles inhaled into the central airways^{4, 5}. Inhaling dust particles could physically harm the alveolar

walls and bronchial epithelial cells⁴. The main components of dust particles, such as minerals including silicon dioxide and aluminum oxide, have been suggested to induce intense irritation and inflammation in the murine lung^{6, 7, 8, 9}. In addition, dust particles can carry microorganisms, such as bacteria, fungi, and viruses¹⁰. With the mice exposed in vivo to dust particles, Ichinose et al. found that these microbial antigens, such as lipopolysaccharide, adhered to the particles' surfaces could cause an increase in pulmonary eosinophils^{11, 12}. COPD patients usually breathe with a large volume at a slow pace to overcome shortness of breath, resulting in a high concentration of dust particles in central airways, causing more severe irritation and inflammations and even leading to death¹³." (Lines 133-151)

"SDS exposure was significantly associated with mortality from several thrombotic diseases in this study, with ischemic stroke mortality having the leading risk (7.49%; 95% CI: 3.12%, 12.05%). Previous studies also showed a positive but nonsignificant association between SDS events and ischemic stroke using emergency visits and hospital admissions data^{14, 15}. We also observed that SDS events were associated with mortality from high blood pressure-related diseases, including intracerebral hemorrhagic stroke mortality (5.40%; 95% CI: 1.25%, 9.73%) and hypertensive heart disease mortality (4.05%; 95% CI: 0.41%, 7.83%). Yang et al.¹⁶ observed a 15% increase in primary intracerebral hemorrhagic stroke admissions associated with SDS event exposure at lag 3. The systemic oxidative stress and inflammation induced by the inhalation of dust particles can cause vascular endothelial damage, increased platelet activity, and enhanced coagulation, thereby promoting thrombus formation^{17, 18}. There is also evidence that high concentrations of dust particles can cause an increase in heart rate and blood pressure^{19, 20}, suggesting the plausibility of the associations of SDS events with stroke mortality and hypertensive heart disease mortality found in our study." (Lines 162-177)

"Existing epidemiological studies have not examined the effects of SDS events on mortality due to genitourinary, nervous, and digestive system diseases. Our study first

found positive associations between SDS events and these mortality outcomes, though not statistically significant. Several epidemiological and experimental studies have investigated the impacts of SDS on genitourinary, nervous, and digestive health. For example, Herrera-Molina et al. found that exposure to SDS events was associated with an increased risk of hospitalizations from genitourinary diseases²¹; a cohort study from Barcelona, Spain, reported a significant increase in gestational age following the Saharan dust episodes²². By collecting blood and urine samples from people affected by the dust storm, Badeenezhad et al. measured biomarkers related to the central nervous system and found that PM₁₀ during SDS could cause neuron and astrocyte damage, leading to neuropsychiatric disorders²³. Cao et al. found that repeated exposure to fine dust particles could cause pathological changes in the stomach of rats²⁴. Our study results and previous findings provide evidence suggesting the potentially harmful impact of SDS events on mortality due to genitourinary, nervous, and digestive system diseases.” (Lines 179-194)

“The differences of health effects between SDS events and conventional PM pollution events (that is, PM_{2.5} pollution events which happen on non-SDS event days) were confirmed by our results. For ischemic stroke, intracerebral hemorrhagic stroke, hypertensive heart disease, chronic lower respiratory disease and COPD, SDS events could trigger more serious impact than conventional PM pollution events. This may be due to the differences in constituents of PM between SDS events and conventional PM pollution events. Sources of PM during SDS include both anthropogenic and natural sources, which is more complicated¹¹. He et al.²⁵ reported that PM_{2.5} during SDS had greater exacerbating effect on the lung eosinophilia of mice than PM_{2.5} in hazy weather. This could result from the increase bioreactivity of PM_{2.5} during SDS. Ho et al.²⁶ found that significant amounts of suspended dust particles during SDS provided platforms to intermix with chemicals on their surfaces. And these reactions may be an unrecognized source of toxic compositions, enhancing the PM_{2.5} toxicity during SDS. Furthermore, fine dust particles can be elevated into the troposphere and travel thousands of kilometers,²⁷ absorbing airborne pollutants from anthropogenic sources in industrial

areas, microorganisms, and potential allergens, such as pollens, and increasing the adverse health effects of PM_{2.5} during SDS^{10, 28}.” (Lines 196-212)

“When accounting for PM_{2.5} and PM_{2.5-10}, we still observed added effects of SDS events on mortality from ischemic stroke, chronic lower respiratory disease, and COPD. Similarly, Sun *et al.*²⁹ observed that heavy PM_{2.5} pollution events, defined as daily average PM_{2.5} concentration $\geq 75 \mu\text{g}/\text{m}^3$ for at least 3 days, had added effects on the circulatory (0.96%; 95% CI: 0.37%, 1.55%) and respiratory (0.55%; 95% CI: -0.52%, 1.63%) mortality. SDS events have coincided with high levels of PM_{2.5-10} and PM_{2.5}³⁰. Experimental studies found that exposure to sustained high PM_{2.5} concentrations could cause severe damage to multiple organs in mice, including cardiac fibrosis and myocardial hypertrophy^{31, 32}. In addition, SDS may evoke the worry and stress which have been shown to contribute to health symptoms of all body systems, including vasoconstriction and increased blood pressure at a cardiovascular level^{33, 34, 35, 36}.” (Lines 214-224)

References for responses:

1. Tam W. W. S., Wong T. W., Wong A. H., Hui D. S. C. Effect of dust storm events on daily emergency admissions for respiratory diseases. *Respirology* **17**, 143-148 (2012).
2. Vodonos A. *et al.* The impact of desert dust exposures on hospitalizations due to exacerbation of chronic obstructive pulmonary disease. *Air Qual. Atmos. Health* **7**, 433-439 (2014).
3. Lorentzou C. *et al.* Extreme desert dust storms and COPD morbidity on the island of Crete. *Int. J. Chronic Obstruct. Pulm. Dis.* **14**, 1763 (2019).
4. Zhang X., Zhao L., Tong D., Wu G., Dan M., Teng B. A systematic review of global desert dust and associated human health effects. *Atmosphere* **7**, (2016).
5. Hsieh N. H., Liao C. M. Assessing exposure risk for dust storm events-associated lung function decrement in asthmatics and implications for control. *Atmos. Environ.* **68**, 256-264 (2013).

6. Eisenbarth S. C., Colegio O. R., O'Connor W., Sutterwala F. S., Flavell R. A. Crucial role for the Nalp3 inflammasome in the immunostimulatory properties of aluminium adjuvants. *Nature* **453**, 1122-1126 (2008).
7. Ichinose T. *et al.* Effects of asian sand dust, Arizona sand dust, amorphous silica and aluminum oxide on allergic inflammation in the murine lung. *Inhal. Toxicol.* **20**, 685-694 (2008).
8. Dorman D. C. *et al.* Biological responses in rats exposed to cigarette smoke and Middle East sand (dust). *Inhal. Toxicol.* **24**, 109-124 (2012).
9. Ghio A. J. *et al.* Biological effects of desert dust in respiratory epithelial cells and a murine model. *Inhal. Toxicol.* **26**, 299-309 (2014).
10. Hashizume M. *et al.* Health effects of Asian dust: A systematic review and meta-analysis. *Environ. Health. Perspect.* **128**, 66001 (2020).
11. Li G. *et al.* Resurgence of sandstorms complicates China's air pollution situation. *Environ. Sci. Technol.* **55**, 11467-11469 (2021).
12. Ichinose T. *et al.* The effects of microbial materials adhered to Asian sand dust on allergic lung inflammation. *Arch. Environ. Contam. Toxicol.* **55**, 348-357 (2008).
13. Seaton A., MacNee W., Donaldson K., Godden D. Particulate air pollution and acute health effects. *Lancet* **345**, 176-178 (1995).
14. Yang C. Y., Chen Y. S., Chiu H. F., Goggins W. B. Effects of Asian dust storm events on daily stroke admissions in Taipei, Taiwan. *Environ. Res.* **99**, 79-84 (2005).
15. Kamouchi M., Ueda K., Ago T., Nitta H., Kitazono T., Fukuoka Stroke Registry I. Relationship between asian dust and ischemic stroke: A time-stratified case-crossover study. *Stroke* **43**, 3085-3087 (2012).
16. Brook R. D. *et al.* Particulate matter air pollution and cardiovascular disease: An update to the scientific statement from the American Heart Association. *Circulation* **121**, 2331-2378 (2010).
17. Piepoli M. F. *et al.* 2016 European Guidelines on cardiovascular disease prevention in clinical practice: The Sixth Joint Task Force of the European Society of Cardiology and Other Societies on Cardiovascular Disease Prevention in Clinical Practice (constituted by representatives of 10 societies and by invited

- experts)Developed with the special contribution of the European Association for Cardiovascular Prevention & Rehabilitation (EACPR). *Eur. Heart J.* **37**, 2315-2381 (2016).
18. Lipsett M. J., Tsai F. C., Roger L., Woo M., Ostro B. D. Coarse particles and heart rate variability among older adults with coronary artery disease in the Coachella Valley, California. *Environ. Health Perspect.* **114**, 1215-1220 (2006).
 19. Chang C. C., Hwang J. S., Chan C. C., Wang P. Y., Cheng T. J. Effects of concentrated ambient particles on heart rate, blood pressure, and cardiac contractility in spontaneously hypertensive rats during a dust storm event. *Inhal. Toxicol.* **19**, 973-978 (2007).
 20. Herrera-Molina E., Gill T. E., Ibarra-Mejia G., Jeon S. Associations between dust exposure and hospitalizations in El Paso, Texas, USA. *Atmosphere* **12**, (2021).
 21. Dadvand P. *et al.* Saharan dust episodes and pregnancy. *J. environ. Monit.* **13**, 3222-3228 (2011).
 22. Badeenezhad A. *et al.* Investigating the relationship between central nervous system biomarkers and short-term exposure to PM₁₀-bound metals during dust storms. *Atmos. Pollut. Res.* **11**, 2022-2029 (2020).
 23. Cao X. J. *et al.* Effects of dust storm fine particle-inhalation on the respiratory, cardiovascular, endocrine, hematological, and digestive systems of rats. *Chin. Med. J. (Engl.)* **131**, 2482-2485 (2018).
 24. He M. *et al.* Differences in allergic inflammatory responses between urban PM_{2.5} and fine particle derived from desert-dust in murine lungs. *Toxicol. Appl. Pharm.* **297**, 41-55 (2016).
 25. Ho K. F. *et al.* Contributions of local pollution emissions to particle bioreactivity in downwind cities in China during Asian dust periods. *Environ. Pollut.* **245**, 675-683 (2019).
 26. Middleton N., Kang U. Sand and dust storms: Impact mitigation. *Sustainability* **9**, 1053 (2017).
 27. Wu Y., Wen B., Li S., Guo Y. Sand and dust storms in Asia: A call for global cooperation on climate change. *Lancet Planet. Health* **5**, e329-e330 (2021).

28. Sun Y. *et al.* Impact of Heavy PM_{2.5} Pollution Events on Mortality in 250 Chinese Counties. *Environ. Sci Technol.* **56**, 8299-8307 (2022).
29. WHO. WHO global air quality guidelines: particulate matter (PM_{2.5} and PM₁₀), ozone, nitrogen dioxide, sulfur dioxide and carbon monoxide (2021).
30. Li D. *et al.* Multiple organ injury in male C57BL/6J mice exposed to ambient particulate matter in a real-ambient PM exposure system in Shijiazhuang, China. *Environ. Pollut.* **248**, 874-887 (2019).
31. Su X. *et al.* Ambient PM_{2.5} caused cardiac dysfunction through FoxO1-targeted cardiac hypertrophy and macrophage-activated fibrosis in mice. *Chemosphere* **247**, 125881 (2020).
32. Stenlund T., Lidén E., Andersson K., Garvill J., Nordin S. Annoyance and health symptoms and their influencing factors: A population-based air pollution intervention study. *Public Health* **123**, 339-345 (2009).
33. Claeson A. S., Lidén E., Nordin M., Nordin S. The role of perceived pollution and health risk perception in annoyance and health symptoms: A population-based study of odorous air pollution. *Int. Arch. Occup. Environ. health* **86**, 367-374 (2013).
34. Orru K., Nordin S., Harzia H., Orru H. The role of perceived air pollution and health risk perception in health symptoms and disease: A population-based study combined with modelled levels of PM₁₀. *Int. Arch. Occup. Environ. Health* **91**, 581-589 (2018).
35. Kwon H. J., Cho S. H., Chun Y., Lagarde F., Pershagen G. Effects of the Asian dust events on daily mortality in Seoul, Korea. *Environ. Res.* **90**, 1-5 (2002).

Second, some analyses (such as effect modification by age and sex) are not helpful in capturing the main message of the paper, but rather add noise to the overall picture, and are not well motivated.

Response:

Thanks for your valuable suggestion. We have comprehensively modified the statistical analysis of our manuscript, with a focus on estimating the overall and added effects of

SDS events. Analyses of the effect modification of age and sex have been moved to the supplementary material.

Concerning the methods:

1. it is not clear why the authors define a 3 level exposure 0-1-2 variable rather than 2 levels 0-1 (what is the purpose of having a level 2?)

Response:

Thank you for raising this point. The level 2 in our study represents exposure to PM pollution (daily PM_{2.5} concentration $\geq 75 \mu\text{g}/\text{m}^3$) during non-SDS event day. This setting was to distinguish the effect of SDS events, which is also a kind of PM pollution, from the effect PM pollution on non-SDS event days. This analysis is primarily intended to explore whether SDS events need to be managed separately from conventional PM pollution events (that is, PM_{2.5} pollution events which happen on non-SDS event days) in terms of health effects.

Conventional PM pollution event is usually caused by anthropogenic activities, while PM pollution on SDS days can be due to both natural and anthropogenic emissions, resulting in differences of PM constituents from conventional PM pollution events^{1,2}. There may also be corresponding differences in the health effects between them. However, in previous studies of the health effects of SDS events, researchers have neglected to distinguish the health effects between SDS events and conventional PM pollution events. This can lead to classify non-SDS event days with PM pollution as SDS event days or classified them as non-SDS event days, resulting in overestimation or underestimation of the effects on health outcomes during SDS event days compared to non-SDS event days. Therefore, the setting of level 2 can also benefit accurately estimating the health effects of SDS events. To communicate clearly, we have modified relevant text in revised manuscript.

Our results confirmed that the health effects of SDS events did differ from those of PM pollution on non-SDS event days, with mortality risks even higher for ischemic stroke,

intracerebral hemorrhagic stroke, hypertensive heart disease, chronic lower respiratory disease and COPD. This also suggests that SDS events, as a special type of PM pollution, have independent health effects and require specialized management of health risks.

Relevant text in revised Methods section:

“to distinguish the effect of PM pollution on non-SDS days, Z_t was a categorical variable with “1” for an identified SDS event day, “2” for a non-SDS event day with $PM_{2.5}$ pollution (that is, daily $PM_{2.5}$ concentration $\geq 75 \mu g/m^3$ on a non-SDS day), and “0” for a clean day (that is, neither SDS event day nor $PM_{2.5}$ pollution day);” (Lines 316-320)

References for responses:

1. Zuo, P. *et al.* Stable iron isotopic signature reveals multiple sources of magnetic particulate matter in the 2021 Beijing sandstorms. *Environ. Sci. Technol. Lett.* **9**, 299–305 (2022).
2. Li G. *et al.* Resurgence of sandstorms complicates China's air pollution situation. *Environ. Sci. Technol.* **55**, 11467-11469 (2021).

2. the definition of SDS days is totally driven by PM concentrations, and the argument that when PM exceeds the thresholds and the ratio is above 0.4 it should be a dust day is only speculative, not supported by any data. Why didn't the authors use external sources of info, such as atmospheric models?

Response:

Thanks for your professional comment. We agreed with your concerns about our definition of SDS events. While, in fact, we used multiple criteria to define SDS events, among which meeting the criterion decided by official sand–dust weather records was the first condition. Due to that the occurrence dates and places of the major sand–dust weather were recorded at provincial level, the other two criteria, concerning PM_{10} concentrations and $PM_{2.5}/PM_{10}$ concentration ratio, were employed to identify county-

specific sand–dust weather. And the criterion “The daily $PM_{2.5}/PM_{10}$ concentration ratio was <0.4 ” was set based on literature review. A low $PM_{2.5}/PM_{10}$ concentration ratio, is one of the prominent features of sand-dust weather, which is ascribed to the overwhelming contribution from dust particles from long-distance transport, used by multiple studies to identify SDS events^{1–4}. For example, the $PM_{2.5}/PM_{10}$ concentration ratios on SDS days ranged between 0.21~0.32 in Phoenix and 0.31~0.43 in Saguaro, USA during 2001-2008⁴. In China, the $PM_{2.5}/PM_{10}$ concentration ratio was usually lower than 0.4 during SDS⁵. Previous studies have used the ratio threshold of 0.4 to identify SDS and investigate the associations between SDS and circulatory and respiratory diseases in China^{1,2}. Additionally, to examine whether the estimated associations were sensitive to the definition of SDS events, we have also conducted several sensitive analyses by rerunning the regression under alternative SDS definitions with different values of $PM_{2.5}/PM_{10}$ concentration ratio. Specifically, we considered two values of $PM_{2.5}/PM_{10}$ concentration ratio (i.e., 0.35, 0.45), and excluded the criterion of $PM_{2.5}/PM_{10}$ concentration ratio in defining SDS events. Results estimated under these alternative definitions of SDS were generally consistent with those estimated under the primary definition of SDS events in our study.

In the revised manuscript, we have modified relevant text in the Methods section and added detailed description to the supplementary material on the choice of methods for the definition of SDS events.

Relevant text in revised Methods section:

“SDS events definition

We also collected the official sand-dust weather records for our study counties from China’s National Meteorological Center. Since the official sand-dust weather is recorded at the province level, we consider PM_{10} concentration and $PM_{2.5}/PM_{10}$ concentration ratio in the SDS events definition. Specifically, for each study county, an SDS event was defined as a day when: (1) there was an official sand–dust weather record on the day; (2) the daily concentration of PM_{10} was $>50 \mu\text{g}/\text{m}^3$ ^{6–8}, which was

the lowest threshold observed in Huffman et al.'s classification of PM₁₀ during SDS^{6,7}; (3) the daily PM_{2.5}/PM₁₀ concentration ratio was <0.4¹. The PM_{2.5}/PM₁₀ concentration ratio is an important indicator to distinguish sand–dust weather from non-sand–dust weather, as the low ratio is often associated with overwhelming contribution from long-distance transport dust particles⁹.

We also considered four alternative definitions of SDS events by using two different thresholds in the PM_{2.5}/PM₁₀ concentration ratio (0.35 and 0.45) and excluding the PM_{2.5}/PM₁₀ concentration ratio in the SDS definition. More details of the SDS events definition are provided in the Supplements.” (Lines 285-301)

Relevant text in revised Supplementary Materials:

“To date, there is no uniform standard for defining SDS events. Several studies used the local official sand–dust weather records in the study area for SDS events definition^{10–21}. Official sand–dust weather records usually have the advantages of authority and accuracy, but most countries do not have unified sand–dust weather records, and researchers often need to compile records from various data sources themselves²⁰. Moreover, the records often do not meet the research needs for the spatial and temporal resolution of sand–dust weather record information; for example, China’s official sand-dust weather records are only recorded at the provincial level.

Thus, some researchers have set a series of indicators based on the physicochemical characteristics of sand–dust weather, such as anomalies in particle matter concentration and composition, anomalies in meteorological factors and weather records, and developed a series of definition methods for SDS events by combining multiple indicators (Table S1)^{2,22–45}. Among these indicators, PM₁₀ concentration and PM_{2.5}/PM₁₀ concentration ratio are relatively important judgment indicators in existing studies because the data are readily available and can reflect the critical physical characteristics of sand-dust weather.

In this study, considering data accessibility and accuracy of results, a combination of multiple indicators, including PM_{10} concentrations and $PM_{2.5}/PM_{10}$ concentration ratio, based on official sand-dust weather records was used to define SDS events.”

Table S1 Indicators for SDS events definition used in existing literature.

Indicator Dimension	Indicator	Examples	Strengths	Limitations
Anomalies in particulate matter concentration	PM_{10} ^{2,22,24-27,29,31-33,36-38,43,44}	Hourly concentration >15 $\mu\text{g}/\text{m}^3$ or daily average concentration >50 $\mu\text{g}/\text{m}^3$	It is the main physical characteristic of sand-dust weather, and relevant data are available.	There is no uniform standard threshold.
	$PM_{2.5}/PM_{10}$ ^{1-3,5}	$PM_{2.5}/PM_{10}$ concentration ratio < 0.4	It is the main physical characteristic of sand-dust weather, and relevant data are available.	There is no uniform standard threshold.
	AQI (Air Quality Index) ¹	AQI > 100	Relevant data are available.	Not a typical feature of sand-dust weather.
Anomalies in particle matter composition	Crustal elements ⁴	Elements such as Al are at least 3 times higher than the usual average value	It is the main physical characteristic of sand-dust weather.	Poor data accessibility.
	Anthropogenic related components ⁴	As, Zn, Cu, Pb, sulfate, nitrate, organic carbon, low values of elemental carbon	It is the main physical characteristic of sand-dust weather.	Poor data accessibility.
Anomalies in meteorological factors	Visibility ^{22,23,26-28,41}	Ground level visibility <10 km	It is the main physical characteristic of sand-dust weather.	Poor data accessibility.
	Wind	Ground wind	It is the main	Fail to recognize

	speed ^{22,34}	speed > 17m/s	physical characteristic of serious sand–dust weather.	sand–dust weather of low intensity, such as floating dust, sand blowing weather.
Anomalies in weather records	Weather observation records of nearby airports ^{18,22}	Record of aircraft grounding at nearby airports due to sand–dust weather	High accuracy.	Poor data accessibility.
	News Media Record ²⁰	News reporting on the occurrence of sand–dust weather	Relevant data are available.	Variable quality, often used for verification of identification results.

References for responses:

1. Tam, W. W. S., Wong, T. W., Wong, A. H. S. & Hui, D. S. C. Effect of dust storm events on daily emergency admissions for respiratory diseases. *Respirology* **17**, 143–148 (2012).
2. Tam, W. W. S., Wong, T. W., & Wong, A. H. Effect of dust storm events on daily emergency admissions for cardiovascular diseases. *Circ. J.* **76**, 655-660 (2012).
3. Zhao, D., Chen, H., Yu, E. & Luo, T. PM_{2.5} /PM₁₀ Ratios in Eight Economic Regions and Their Relationship with Meteorology in China. *Adv. Meteorol.* **2019**, 1–15 (2019).
4. Tong, D. Q., Wang, J. X. L., Gill, T. E., Lei, H. & Wang, B. Intensified dust storm activity and Valley fever infection in the southwestern United States. *Geophys. Res. Lett.* **44**, 4304–4312 (2017).
5. Filonchyk, M. Characteristics of the severe March 2021 Gobi Desert dust storm and its impact on air pollution in China. *Chemosphere* **287**, 132219 (2022).
6. Hoffmann, C., Funk, R., Wieland, R., Li, Y. & Sommer, M. Effects of grazing and topography on dust flux and deposition in the Xilingele grassland, Inner Mongolia.

- J. Arid Environ.* **72**, 792–807 (2008).
7. Hoffmann, C., Funk, R., Sommer, M. & Li, Y. Temporal variations in PM₁₀ and particle size distribution during Asian dust storms in Inner Mongolia. *Atmos. Environ.* **42**, 8422–8431 (2008).
 8. Khaniabadi, Y. O. *et al.* Hospital admissions in Iran for cardiovascular and respiratory diseases attributed to the Middle Eastern dust storms. *Environ. Sci. Pollut. Res.* **24**, 16860–16868 (2017).
 9. Ma, Y. *et al.* Comparison of inorganic chemical compositions of atmospheric TSP, PM₁₀ and PM_{2.5} in northern and southern Chinese coastal cities. *J. Environ. Sci.* **55**, 339–353 (2017).
 10. Chan, C.-C. & Ng, H.-C. A case-crossover analysis of Asian dust storms and mortality in the downwind areas using 14-year data in Taipei. *Sci. Total Environ.* **410–411**, 47–52 (2011).
 11. Kojima, S. *et al.* Asian dust exposure triggers acute myocardial infarction. *Eur. Heart J.* **38**, 3202–3208 (2017).
 12. Jung, J. *et al.* Burden of dust storms on years of life lost in Seoul, South Korea: A distributed lag analysis. *Environ. Pollut.* **296**, 118710 (2022).
 13. Matsukawa, R. *et al.* Desert dust is a risk factor for the incidence of acute myocardial infarction in western Japan. *Circ. Cardiovasc. Qual. Outcomes* **7**, 743–748 (2014).
 14. Meng, Z. & Lu, B. Dust events as a risk factor for daily hospitalization for respiratory and cardiovascular diseases in Minqin, China. *Atmos. Environ.* **41**, 7048–7058 (2007).
 15. Lee, H., Kim, H., Honda, Y., Lim, Y.-H. & Yi, S. Effect of Asian dust storms on daily mortality in seven metropolitan cities of Korea. *Atmos. Environ.* **79**, 510–517 (2013).
 16. Kwon, H.-J., Cho, S.-H., Chun, Y., Lagarde, F. & Pershagen, G. Effects of the Asian dust events on daily mortality in Seoul, Korea. *Environ. Res.* **90**, 1–5 (2002).
 17. Wang, Y.-C. & Lin, Y.-K. Mortality associated with particulate concentration and Asian dust storms in Metropolitan Taipei. *Atmos. Environ.* **117**, 32–40 (2015).

18. Neophytou, A. M. *et al.* Particulate matter concentrations during desert dust outbreaks and daily mortality in Nicosia, Cyprus. *J. Expo. Sci. Environ. Epidemiol.* **23**, 275–280 (2013).
19. Kamouchi, M. *et al.* Relationship between Asian dust and ischemic stroke: A time-stratified case-crossover study. *Stroke* **43**, 3085–3087 (2012).
20. Crooks, J. L. *et al.* The association between dust storms and daily non-accidental mortality in the United States, 1993–2005. *Environ. Health Persp.* **124**, 1735–1743 (2016).
21. Kim, H.-S., Kim, D.-S., Kim, H. & Yi, S.-M. Relationship between mortality and fine particles during Asian dust, smog–Asian dust, and smog days in Korea. *Int. J. Environ. Health Res.* **22**, 518–530 (2012).
22. Middleton, N. *et al.* A 10-year time-series analysis of respiratory and cardiovascular morbidity in Nicosia, Cyprus: The effect of short-term changes in air pollution and dust storms. *Environ. Health* **7**, 39 (2008).
23. Ishii, M. *et al.* Association of short term exposure to Asian dust with increased blood pressure. *Sci. Rep.* **10**, 17630 (2020).
24. Stafoggia, M. *et al.* Desert dust outbreaks in southern Europe: Contribution to daily PM₁₀ concentrations and short-term associations with mortality and hospital admissions. *Environ. Health Persp.* **124**, 413–419 (2016).
25. Samoli, E., Kougea, E., Kassomenos, P., Analitis, A. & Katsouyanni, K. Does the presence of desert dust modify the effect of PM₁₀ on mortality in Athens, Greece? *Sci. Total Environ.* **409**, 2049–2054 (2011).
26. Lee, H. *et al.* Effect of Asian dust storms on mortality in three Asian cities. *Atmos. Environ.* **89**, 309–317 (2014).
27. Aghababaeian, H. *et al.* Effect of dust storms on non-accidental, cardiovascular, and respiratory mortality: A case of dezfoul city in Iran. *Environ. Health Insights* **15**, 11786302211060152 (2021).
28. Lee, S. *et al.* Effects of Asian dust-derived particulate matter on ST-elevation myocardial infarction: Retrospective, time series study. *BMC Public Health* **21**, 68 (2021).

29. Johnston, F., Hanigan, I., Henderson, S., Morgan, G. & Bowman, D. Extreme air pollution events from bushfires and dust storms and their association with mortality in Sydney, Australia 1994–2007. *Environ. Res.* **111**, 811–816 (2011).
30. Zhang, Q., Zhang, J., Yang, Z., Zhang, Y. & Meng, Z. Impact of PM_{2.5} derived from dust events on daily outpatient numbers for respiratory and cardiovascular diseases in Wuwei, China. *Procedia Environ. Sci.* **18**, 290–298 (2013).
31. Domínguez-Rodríguez, A. *et al.* Impact of Saharan dust on the incidence of acute coronary syndrome. *Rev. Esp. Cardiol.* **74**, 321–328 (2021).
32. Vodonos, A. *et al.* Individual effect modifiers of dust exposure effect on cardiovascular morbidity. *PLoS One* **10**, e0137714 (2015).
33. Teng, J. C.-Y., Chan, Y.-S., Peng, Y.-I. & Liu, T.-C. Influence of Asian dust storms on daily acute myocardial infarction hospital admissions. *Public Health Nurs.* **33**, 118–128 (2016).
34. Ho, H. C., Wong, M. S., Yang, L., Chan, T.-C. & Bilal, M. Influences of socioeconomic vulnerability and intra-urban air pollution exposure on short-term mortality during extreme dust events. *Environ. Pollut.* **235**, 155–162 (2018).
35. Jiménez, E., Linares, C., Martínez, D. & Díaz, J. Role of Saharan dust in the relationship between particulate matter and short-term daily mortality among the elderly in Madrid (Spain). *Sci. Total Environ.* **408**, 5729–5736 (2010).
36. Zauli Sajani, S. *et al.* Saharan dust and daily mortality in Emilia-Romagna (Italy). *Occup. Environ. Med.* **68**, 446–451 (2011).
37. Perez, L. *et al.* Saharan dust, particulate matter and cause-specific mortality: A case–crossover study in Barcelona (Spain). *Environ. Int.* **48**, 150–155 (2012).
38. Al-Taiar, A. & Thalib, L. Short-term effect of dust storms on the risk of mortality due to respiratory, cardiovascular and all-causes in Kuwait. *Int. J. Biometeorol.* **58**, 69–77 (2014).
39. Ma, Y. *et al.* Short-term effects of air pollution on daily hospital admissions for cardiovascular diseases in western China. *Environ. Sci. Pollut. Res.* **24**, 14071–14079 (2017).
40. Renzi, M. *et al.* Short-term effects of desert and non-desert PM₁₀ on mortality in

- Sicily, Italy. *Environ. Int.* **120**, 472–479 (2018).
41. JROAD Investigators *et al.* Short-term exposure to desert dust and the risk of acute myocardial infarction in Japan: A time-stratified case-crossover study. *Eur. J. Epidemiol.* **35**, 455–464 (2020).
 42. Hwang, S. S. *et al.* The Asian dust events and hospital admissions with respiratory and cardiovascular disease in Seoul, Korea: Isee-249. *Epidemiology* **14**, S48 (2003).
 43. Byun, G., Kim, H., Choi, Y. & Lee, J.-T. The difference in effect of ambient particles on mortality between days with and without yellow dust events: Using a larger dataset in Seoul, Korea from 1998 to 2015. *Sci. Total Environ.* **691**, 819–826 (2019).
 44. Bell, M. L., Levy, J. K. & Lin, Z. The effect of sandstorms and air pollution on cause-specific hospital admissions in Taipei, Taiwan. *Occup. Environ. Med.* **65**, 104–111 (2008).
 45. Tchounwou, P. The effects of PM_{2.5} from Asian dust storms on emergency room visits for cardiovascular and respiratory diseases. *Int. J. Environ. Res. Public Health* **1**, 1–2 (2004).

3. some choices (number of DF for time trends or temperature, lag terms for temperature) are questionable and not adequately motivated (despite the sensitivity analyses)

Response:

Thanks for your professional comment. We made the choice of df for time trends and temperature based on previous epidemiologic studies on climate extreme weather events^{1–8}. Climate extremes events, such as heat waves, cold spells, and tropical cyclones, typically occur within specific seasons. These previous studies often limit the analysis to the specific periods (i.e., the warm or cold season) to eliminate any confounding bias that could arise from factors varying across the whole year in a computationally efficient way, with 2 df typically used for time trends and 3 df typically used for weather variables. SDS event is also a kind of extreme weather event. Thus, we analyzed the health effects of SDS events using data for the SDS period (1 February

- 31 May) rather than the whole year. By referring to the experience of previous studies, 2 df for time trends and 3 df for temperature were applied in our study. We also used 3 df per SDS period for time trends, 4 df and 5 df for temperature in the sensitivity analysis, with results being consistent with our primary results.

The choice of lag terms for temperature was made based on previous epidemiologic studies on the short-term effects of air pollution^{9,10}. These study usually adjusted data for temperature and relative humidity with the same lagged day as air pollution exposure in the regression models when estimating the lagged mortality risks of air pollution. To communicate clearly, we have also modified relevant text to provide a more precise description of lag terms. In addition, there are researches considering the long delay of effects of weather variables by using 21-day moving average of temperature and the 7-day moving average of relative humidity¹¹. Thus, we also added new sensitivity analysis with this setting. The results corroborate the robustness of our results.

Relevant text in revised Methods section:

“In addition, to investigate the potential delayed effects of SDS events exposure, we performed lagged analysis by fitting the same model separately for three single-day lagged data (lag 1, 2, and 3). For example, analysis at lag 1 estimates the impact on mortality on day t (Y_t) associated with exposure to the previous day (i.e., Z_{t-1}). In the lagged analysis, we used data for the same lagged day for $Temp_t$ and RH_t .” (Lines 342-346)

“Second, we changed the df for the time trend variable ($df = 3$) and used two different df ($df = 4, 5$) for meteorological parameters in the spline functions. Third, instead of the daily mean temperature and relative humidity, we used the 21-day moving average of temperature and the 7-day moving average of relative humidity to fully adjust for the confounding of meteorological conditions¹¹.” (Lines 352-357)

Relevant text in revised Results section:

“Results from sensitivity analyses, by changing the degree of freedom of spline functions, changing the adjustment of meteorological parameters (Fig. S6), and using the data of different study periods (Fig. S7), generally remained consistent with those from the main models for most mortality outcomes.” (Lines 104-108)

Fig. S6. Excess risk (ER, %) and 95% confidence interval for mortality associated with sand and dust storms events using different model settings. “Model 1” represents estimates from the primary model, with the degree of freedom (df) of 2 and 3, for time variable and meteorological parameters in natural spline functions. “Model 2” represents estimates from the primary model, except the df for the time variable was 3. “Model 3” represents estimates from the primary model, except the df for the time variable was 3, and the dfs for meteorological parameters were 4. “Model 4” represents estimates from the primary model, except the df for the time variable was 3, and the dfs for meteorological parameters were 5. “Model 5” represents estimates from the primary model, except using a 21-day moving average of temperature and a 7-day moving average of relative humidity. Red represents that the ER differed significantly from 0% ($P < 0.05$). The Mortality of broad causes

results are shown in the top two panels, and mortality of specific causes results in the bottom two panels.

References for responses:

1. Wang, Q. *et al.* Independent and combined effects of heatwaves and PM_{2.5} on preterm birth in Guangzhou, China: A survival analysis. *Environ. Health Persp.* **128**, 017006 (2020).
2. Sun, Z. *et al.* Heat wave characteristics, mortality and effect modification by temperature zones: A time-series study in 130 counties of China. *Int. J. Epidemiol.* **49**, 1813–1822 (2021).
3. Zhao, Q. *et al.* The association between heatwaves and risk of hospitalization in Brazil: A nationwide time series study between 2000 and 2015. *PLoS Med.* **16**, e1002753 (2019).
4. Guo, Y. *et al.* Quantifying excess deaths related to heatwaves under climate change scenarios: A multicountry time series modelling study. *PLoS Med.* **15**, e1002629 (2018).
5. Anderson, G. B. & Bell, M. L. Heat waves in the United States: Mortality risk during heat waves and effect modification by heat wave characteristics in 43 U.S. communities. *Environ. Health Persp.* **119**, 210–218 (2011).
6. Hansen, A. *et al.* The effect of heat waves on mental health in a temperate Australian city. *Environ. Health Persp.* **116**, 1369–1375 (2008).
7. Ma, C. *et al.* Cold spells and cause-specific mortality in 47 Japanese prefectures: A systematic evaluation. *Environ. Health Persp.* **129**, 067001.
8. Hansen, A. L. *et al.* The effect of heat waves on hospital admissions for renal disease in a temperate city of Australia. *Int. J. Epidemiol.* **37**, 1359–1365 (2008).
9. Ban, J., Su, W., Zhong, Y., Liu, C. & Li, T. Ambient formaldehyde and mortality: A time series analysis in China. *Sci. Adv.* **8**, eabm4097 (2022).
10. Sun, Y. *et al.* Impact of heavy PM_{2.5} pollution events on mortality in 250 Chinese counties. *Environ. Sci. Technol.* **56**, 8299–8307 (2022).
11. Ye, T. *et al.* Risk and burden of hospital admissions associated with wildfire-

related PM_{2.5} in Brazil, 2000–15: A nationwide time-series study. *Lancet Planet. Health*, **5**, e599-e607 (2021).

4. the presumed "contribution" of PM_{2.5} and PM_{2.5-10} is purely speculative: what the authors estimate is the effect of SDS 0/1 with or without adjustment for PM. The difference between such estimates does not say anything on the role of PM on cause-specific mortality during SDS days,

Response:

Thanks for your professional comment. We agree that our original analytical approach (with or without adjustment for PM) did not investigate the role of PM on the mortality effects of SDS. Instead, effect estimates from the model with no PM adjusted represent the overall effects of SDS events, and effect estimates from the model with PM adjusted represent the added effects of SDS events. Therefore, we have modified the relevant text in the revised manuscript.

Relevant text in revised Introduction section:

“Inhalable particulate matter (PM₁₀), constituted of fine particulate matter (PM_{2.5}) and coarse fine particulate matter (PM_{2.5-10}), is well known to be the main component of SDS, posing a threat to human health¹. For example, Neophytou, and colleagues observed that a 10 µg/m³ increase in PM₁₀ concentration during SDS was associated with a 2.43% increase in cardiovascular mortality². Moreover, there is evidence that heavy PM_{2.5} pollution events increased mortality risks and caused an independently added effect³. Whether SDS events, a special kind of particulate matter (PM) pollution event with high concentration of PM_{2.5-10} and PM_{2.5}, have added effects on mortality is yet to be determined.

Here, we conducted a nationwide multicenter time series study in China. Our objectives were to: (1) investigate the overall short-term effects of SDS events on mortality from a series of causes, identifying the spectrum of SDS-sensitive health outcome; (2) explore the added short-term effects of SDS events on mortality. Findings from this study will

improve current understanding of the health effects of SDS.” (Lines 38-52)

Relevant text in revised Methods section:

“Study design

We first collected data for counties frequently affected by SDS; the distribution of study counties was designed to cover the primary SDS transmission routes, with a good representation of the heterogeneity in exposure levels to SDS events (Fig. 1). We then performed a two-stage time series analysis using the daily data from 2013 to 2018 for 214 Chinese counties. Further, we investigated the added effects of SDS events by controlling $PM_{2.5-10}$ and $PM_{2.5}$ in the models, respectively. Fig. S8 shows a diagram of our study design.” (Lines 253-260)

“Since SDS events are typically PM pollution, we further investigate whether SDS events exposure had added effects on mortality by adjusting for $PM_{2.5-10}$ and $PM_{2.5}$ in the GLM models.” (Lines 340-342)

Relevant text in revised Results section:

“With adjustments of $PM_{2.5-10}$ and $PM_{2.5}$ in the regression models, the added effects of SDS events on mortality were generally less substantial than the overall effects of SDS events (Fig. 3). Despite this, we still observed significantly increased risk associated with SDS events for ischemic stroke, chronic lower respiratory disease, and COPD, suggesting added health effects of SDS events on these mortality outcomes (Fig. 3).” (Lines 88-93)

Fig. 3. Excess risk (ER, %) and 95% confidence interval (CI) for mortality due to SDS-sensitive health outcomes. Results are shown for mortality risks associated with sand and dust storms from main analysis (“Main effect”), (“Controlling PM_{2.5-10}”), and (“Controlling PM_{2.5}”). ISTR: ischemic stroke mortality; IHDSTR: intracerebral hemorrhagic stroke mortality; HBP: hypertensive heart disease mortality; MI: myocardial infarction mortality; AMI: acute myocardial infarction mortality; AIHD: acute ischemic heart disease mortality; CLRI: chronic lower respiratory disease mortality; COPD: chronic obstructive pulmonary disease mortality.

Relevant text in revised Discussion section:

“When accounting for PM_{2.5} and PM_{2.5-10}, we still observed added effects of SDS events on mortality from ischemic stroke, chronic lower respiratory disease, and COPD. Similarly, Sun et al.³ observed that heavy PM_{2.5} pollution events, defined as daily average PM_{2.5} concentration $\geq 75 \mu\text{g}/\text{m}^3$ for at least 3 days, had added effects on the circulatory (0.96%; 95% CI: 0.37%, 1.55%) and respiratory (0.55%; 95% CI: -0.52%, 1.63%) mortality. SDS events have coincided with high levels of PM_{2.5-10} and PM_{2.5}⁴. Experimental studies found that exposure to sustained high PM_{2.5} concentrations could cause severe damage to multiple organs in mice, including cardiac fibrosis and myocardial hypertrophy^{5, 6}. In addition, SDS may evoke the worry and stress which have been shown to contribute to health symptoms of all body systems, including vasoconstriction and increased blood pressure at a cardiovascular level^{7, 8, 9, 10}.” (Lines 214-224)

References for responses:

1. Wu Y., Wen B., Li S., Guo Y. Sand and dust storms in Asia: A call for global cooperation on climate change. *Lancet Planet. Health* **5**, e329-e330 (2021).
2. Neophytou A. M. *et al.* Particulate matter concentrations during desert dust outbreaks and daily mortality in Nicosia, Cyprus. *J. Expo. Sci. Environ. Epidemiol.* **23**, 275-280 (2013).
3. Sun Y. *et al.* Impact of heavy PM_{2.5} pollution events on mortality in 250 Chinese counties. *Environ. Sci. Technol.* **56**, 8299-8307 (2022).
4. WHO. WHO global air quality guidelines: particulate matter (PM_{2.5} and PM₁₀), ozone, nitrogen dioxide, sulfur dioxide and carbon monoxide (2021).
5. Li D. *et al.* Multiple organ injury in male C57BL/6J mice exposed to ambient particulate matter in a real-ambient PM exposure system in Shijiazhuang, China. *Environ. Pollut.* **248**, 874-887 (2019).
6. Su X. *et al.* Ambient PM_{2.5} caused cardiac dysfunction through FoxO1-targeted cardiac hypertrophy and macrophage-activated fibrosis in mice. *Chemosphere* **247**, 125881 (2020).
7. Stenlund T., Lidén E., Andersson K., Garvill J., Nordin S. Annoyance and health symptoms and their influencing factors: A population-based air pollution intervention study. *Public Health* **123**, 339-345 (2009).
8. Claeson A. S., Lidén E., Nordin M., Nordin S. The role of perceived pollution and health risk perception in annoyance and health symptoms: A population-based study of odorous air pollution. *Int. arch. occup. environ. health* **86**, 367-374 (2013).
9. Orru K., Nordin S., Harzia H., Orru H. The role of perceived air pollution and health risk perception in health symptoms and disease: A population-based study combined with modelled levels of PM₁₀. *Int. Arch. Occup. Environ. Health* **91**, 581-589 (2018).
10. Kwon H. J., Cho S. H., Chun Y., Lagarde F., Pershagen G. Effects of the Asian dust events on daily mortality in Seoul, Korea. *Environ. Res.* **90**, 1-5 (2002).

As a consequence, the interpretation of the results looks overstated, and the general picture is chaotic.

Response:

Thanks again for your professional and constructive suggestions. We agree our original manuscript needed to interpret the results and describe the contribution of our findings to the existing literature. We have thoroughly revised the manuscript with a point-by-point response provided above. We hope you will be more enthusiastic about our manuscript after we have addressed these issues from your comments. Our main findings are summarized below:

With nationwide mortality data for many causes, our study, for the first time, reported positive associations of SDS events with mortality due to a series of cardiopulmonary sub-causes, including ischemic stroke, intracerebral hemorrhagic stroke, hypertensive heart disease, myocardial infarction, acute myocardial infarction, acute ischemic heart disease, chronic lower respiratory disease, and COPD. We also examined the effects of SDS events on mortality from genitourinary, nervous, and digestive system diseases, which have not been reported in previous epidemiologic studies. Further, we found significant added effects of SDS events for ischemic stroke, chronic lower respiratory disease, and COPD mortality. These findings could provide scientific evidence to deepen the current understanding of SDS's health effects and to plan interventions to protect the public against SDS.

Relevant text in revised Discussion section:

“To the best of our knowledge, this study is the first to elucidate the mortality risks of SDS using a large sample size and a spectrum of mortality outcomes. Respiratory mortality significantly and substantially increased during SDS event days compared with clean days (8.90%; 95% CI: 4.96%, 12.98%). We identified a spectrum of SDS-sensitive health outcomes, including ischemic stroke mortality, intracerebral hemorrhagic stroke mortality, hypertensive heart disease mortality, myocardial

infarction mortality, acute myocardial infarction mortality, acute ischemic heart disease mortality, chronic lower respiratory disease mortality, and COPD mortality. Added effects of SDS events were observed for mortality due to ischemic stroke, chronic lower respiratory disease, and COPD. Findings from this study provided scientific evidence to deepen the current understanding of SDS's health effects and to plan interventions to protect the public against SDS.” (Lines 113-125)

REVIEWERS' COMMENTS:

Reviewer #1 (Remarks to the Author):

The authors have adequately addressed my comments.

Reviewer #2 (Remarks to the Author):

The authors have adequately addressed most of my concerns. The very small daily cases of some death causes could result in considerable uncertainty to the results. The argument need to be more thoroughly validated rather than only referring to the literature. And this point should be also added as a limitation.

Reviewer #3 (Remarks to the Author):

The authors made an excellent job in responding to all major and minor concerns. Some of the issues I have previously raised remain partially unanswered (using atmospheric models to identify SDS events, adding clarity in the relative role of SDS and PM, expanding the causes of death), but at this stage, and with the data in their hands, there is not much else the authors can do.

My only final suggestion is to add a few lines about non-CVD and non-RESP causes of death: as I previously said, much has been already demonstrated on the harmful effects of SDS on CVD and RESP mortality, while little is known on other causes. Even if the study is inconclusive on such other causes, I think the authors can stress a little bit more the importance of investigating new causes of death in relation to desert dust exposure.

Responses to reviewers' comments for "Mortality risks from a spectrum of causes associated with sand and dust storms in China"

Thanks, the editor and reviewers for acknowledging the worth of the present manuscript, and the thoughtful and constructive suggestions. We have responded point-by-point to the reviewer's comments below.

Responses to Reviewer #1:

The authors have adequately addressed my comments.

Response:

Thank you. We appreciate your constructive and helpful comments in the previous review.

Responses to Reviewer #2:

The authors have adequately addressed most of my concerns. The very small daily cases of some death causes could result in considerable uncertainty to the results. The argument needs to be more thoroughly validated rather than only referring to the literature. And this point should be also added as a limitation.

Response:

Thank you for your professional suggestions. Yes, the small daily county-level cases of some mortality outcomes may influence the statistical power and introduce uncertainty. However, from our team's experience on county-level mortality data analysis, the number of cases of death >100 per year for four years can provide sufficient statistical validity. And the county-level mortality data in this study merits a long time series (6 years) and enough counties (214 counties). Moreover, in the current revision, we added a sensitivity analysis to investigate the robustness of our results due to this issue. We rerun our models by excluding counties with an average daily death count lower than one during the SDS period, 2013-2018. The results were generally consistent with our main results. Thus, we believe that the present results are plausible; and the situation, that the daily county-level cases of some death causes are very small, is not sufficient to be considered a limitation of our study. Relevant text about the sensitivity analysis

was added in the Methods and Results sections.

Relevant text in revised Methods section:

“Fifth, since the daily county-level death counts were pretty small for certain mortality outcomes, we conducted sensitivity analyses only on study counties with daily death counts exceeding one during the SDS periods. This approach allowed us to examine the potential uncertainty introduced by these low counts.” (Lines 372-376)

Relevant text in revised Results section:

“Results from sensitivity analyses, by changing the degree of freedom of spline functions, changing the adjustment of meteorological parameters (Fig. S6), using the data of different study periods (Fig. S7), and using the data of different study counties (Fig. S8), generally remained consistent with those from the main models for most mortality outcomes.” (Lines 119-123)

Fig. S8. Mortality risk associated with sand and dust storms (SDS) events based on models fit using data of different study counties. “All counties” represents estimates

from our primary analysis using data for all study counties. “Subset counties” represents estimates from the sensitivity analysis excluding counties with an average daily death count of less than one during the SDS period (1 February–31 May), 2013–2018. The number of counties used in the sensitivity analysis ranged from 44 to 194. Points represent the estimated excess risk (ER, %). Horizontal lines represent the 95% confidence interval (CI). Red represents that the ER differed significantly from 0% ($P < 0.05$). The Mortality of broad causes results are shown in the top two panels, and mortality of specific causes results in the bottom two panels.

Reviewer #3 (Remarks to the Author):

The authors made an excellent job in responding to all major and minor concerns. Some of the issues I have previously raised remain partially unanswered (using atmospheric models to identify SDS events, adding clarity in the relative role of SDS and PM, expanding the causes of death), but at this stage, and with the data in their hands, there is not much else the authors can do.

My only final suggestion is to add a few lines about non-CVD and non-RESP causes of death: as I previously said, much has been already demonstrated on the harmful effects of SDS on CVD and RESP mortality, while little is known on other causes. Even if the study is inconclusive on such other causes, I think the authors can stress a little bit more the importance of investigating new causes of death in relation to desert dust exposure.

Response:

We appreciate your professional suggestions. We added some text to discuss the importance of studying the effects of SDS on other mortality outcomes in addition to cardiorespiratory diseases.

Relevant text in revised Discussion section:

“Our study first examined the effects of SDS events on mortality due to genitourinary, nervous, and digestive system diseases; we found positive associations between SDS events and these mortality outcomes, though not statistically significant. These were

consistent with several epidemiological studies focusing on morbidity and experimental studies. For example, Herrera-Molina et al. found that exposure to SDS events was associated with an increased risk of hospitalizations from genitourinary diseases¹. By collecting blood and urine samples from people affected by the dust storm, Badeenezhad et al. measured biomarkers related to the central nervous system and found that PM₁₀ during SDS could cause neuron and astrocyte damage, leading to neuropsychiatric disorders². Cao et al. found that repeated exposure to fine dust particles could cause pathological changes in the stomach of rats³. Our study results and previous findings provide evidence suggesting the potentially harmful impact of SDS events on mortality due to genitourinary, nervous, and digestive system diseases. More research is needed to investigate the effects of SDS on other diseases in addition to cardiorespiratory diseases.” (Lines 194-208)

References for responses:

1. Herrera-Molina E., Gill T. E., Ibarra-Mejia G., Jeon S. Associations between dust exposure and hospitalizations in El Paso, Texas, USA. *Atmosphere* **12**, 1413 (2021).
2. Badeenezhad A. et al. Investigating the relationship between central nervous system biomarkers and short-term exposure to PM₁₀-bound metals during dust storms. *Atmos. Pollut. Res.* **11**, 2022-2029 (2020).
3. Cao X. J. et al. Effects of dust storm fine particle-inhalation on the respiratory, cardiovascular, endocrine, hematological, and digestive systems of rats. *Chin. Med. J. (Engl.)* **131**, 2482-2485 (2018).